# Cell shape anisotropy contributes to self-organized feather pattern fidelity in birds

**Camille Curantz[1,2,3], Richard Bailleul[1,2,4], María Castro-Scherianz[1], Magdalena Hidalgo[1], Melina Durande[5], François Graner[5], Marie Manceau[1] ***

**1** Center for Interdisciplinary Research in Biology, CNRS UMR7241, INSERM U1050, Collège de France, Paris, France, **2** Sorbonne University, UPMC Paris VI, Paris, France, **3** Centre for Chromosome Biology, National University of Ireland Galway, Galway, Ireland, **4** Developmental Biology & Cell Biology and Biophysics Units, European Molecular Biology Laboratory, Heidelberg, Germany, **5** Matière et Systèmes Complexes, Université de Paris, CNRS UMR 7057, Paris, France

* marie.manceau@college-de-france.fr

**Data Availability Statement:** All relevant data are within the paper and its Supporting Information files. The data underlying figures can be found at 10.5281/zenodo.7006365.

## Abstract

Developing tissues can self-organize into a variety of patterned structures through the stabilization of stochastic fluctuations in their molecular and cellular properties. While molecular factors and cell dynamics contributing to self-organization have been identified in vivo, events channeling self-organized systems such that they achieve stable pattern outcomes remain unknown. Here, we described natural variation in the fidelity of self-organized arrays formed by feather follicle precursors in bird embryos. By surveying skin cells prior to and during tissue self-organization and performing species-specific ex vivo drug treatments and mechanical stress tests, we demonstrated that pattern fidelity depends on the initial amplitude of cell anisotropy in regions of the developing dermis competent to produce a pattern. Using live imaging, we showed that cell shape anisotropy is associated with a limited increase in cell motility for sharp and precisely located primordia formation, and thus, proper pattern geometry. These results evidence a mechanism through which initial tissue properties ensure stability in self-organization and thus, reproducible pattern production.

## Introduction

Self-organization is a pattern-forming process by which spatial arrangement spontaneously emerges from the amplification and stabilization of small-scale stochastic fluctuations [1–4]. This process is thought to produce numerous animal patterns at different scales, from whole populations (e.g., flocks and swarms) or individuals (e.g., color patterns) to organs (e.g., finger prints, digits) and subcellular structures (e.g., cytoskeleton networks; for review [5]). Geometries obtained through self-organized patterning range from trees, spirals, meanders, and waves, to tessellations, cracks, stripes, or spots [5]. The nature of intrinsic, fluctuating tissue properties that trigger and contribute to self-organization in biological systems has been long sought after, and several studies evidenced the key roles of diffusing molecular factors such as morphogens [6–10], emergent cellular processes such as coordinated movement [11], material

**Funding:** This work was funded by an European
Research Council (ERC) Starting Grant (#639060)
and a Paris Sciences et Lettres (PSL) University
Grant to MM. The funders had no role in study
design, data collection and analysis, decision to
publish, or preparation of the manuscript.

**Competing interests:** The authors have declared
that no competing interests exist.

properties such as extracellular matrix (ECM)-dependent rigidity [12] and viscosity [13], and
mechanical events involving cell-driven force generation and transmission [11,13–16].

Self-organization per se is robust to perturbations, but its outcomes are highly sensitive to
the initial conditions of the biological system (e.g., parameters defining the naïve state of the
un-patterned population, tissue, or cell): Patterns will typically emerge regardless of changes in
system properties, but their type or geometry may be drastically modified [1–5]. The inherent
malleability of self-organization is compatible with the extensive diversity in animal's natural
patterns, which sometimes arises rapidly on an evolutionary scale. A pattern may transform
from one geometry to another through minor changes in the biological parameters of a pat-
terning factor (e.g., concentration, diffusivity) that involved a limited number of genetic modi-
fications [17]. However, self-organization alone hardly reconciles with the meticulous
precision at which patterns are produced in individuals of a species, a fidelity necessary to
guarantee survival and reproductive success [18]. A pressing problem is thus to identify mech-
anisms through which fluctuations are channeled such that self-organized developmental sys-
tems achieve pattern fidelity. Theoretical work based on models of self-organized behavioral
or developmental patterns predicted that these mechanisms involve initial intrinsic properties
of the naïve system [19–22], but such properties have not been experimentally explored, and
the nature of events shaping precision in tissue response—and therefore in pattern geometries,
has remained a black box.

## Results

### Self-organized feather follicle pattern geometry varies between avian species

To uncover the embryonic properties controlling self-organized pattern fidelity, we studied
the spatial distribution of feather follicle precursors in geometrical arrays in the avian skin (Fig
1A; [23,24]). During development, the skin divides in regions of follicle-forming competence
marked by higher expression of *ß-catenin* transcripts [25,26] (i.e., competence stage). At that
stage, the ß-catenin protein locates at the membrane of cells in the overlying epithelial layer of
the skin, the epidermis (Fig 1B). Competent skin regions self-organize: local condensations of
cells in the epidermis and its underlying mesenchymal layer, the dermis, form follicle precur-
sors, or primordia [12,23–26], that can be visualized by DAPI/phalloïdin stains marking cell
nuclei and peripheral actin (i.e., condensation stage; Fig 1C). Together, nascent primordia cre-
ate a "dotted pattern." Numerical and empirical work suggested that the self-organized emer-
gence of primordia patterns is driven by reaction-diffusion and chemotaxis dynamics that
involve molecular factors of the BMP, FGF, and Wnt signaling pathways [25–33] and by
mechanical properties of the developing skin tissue such as calcium signaling-dependent con-
tractility of dermal mesenchymal cells [13,34]. In each primordium, the mechanically induced
translocation of the ß-catenin protein from the membrane to the nucleus of epidermal cells
marks the onset of differentiation into a feather follicle [12] (i.e., differentiation stage; Fig 1D).

Feather primordia pattern geometries are typical to each bird group [23,26,33], indicating
that primordia self-organization is constrained by processes ensuring species-specific pattern
fidelity. To identify these processes, we first compared dynamics of primordia pattern emer-
gence between 6 avian species from the 3 main clades of the bird phylogeny (Fig 2A). In the
zebra finch *Taeniopygia guttata*, part of the species-rich clade *Neoaves*, and in the domestic
chicken *Gallus gallus* and Japanese quail *Coturnix japonica*, representing the *Galloanserae*
clade, we observed that the initial competent skin regions expressing *ß-catenin* transcripts are
longitudinal segments. These segments are located bilaterally to the midline in the zebra finch
and Japanese quail and medially in the domestic chicken. Primordia emerged along segments

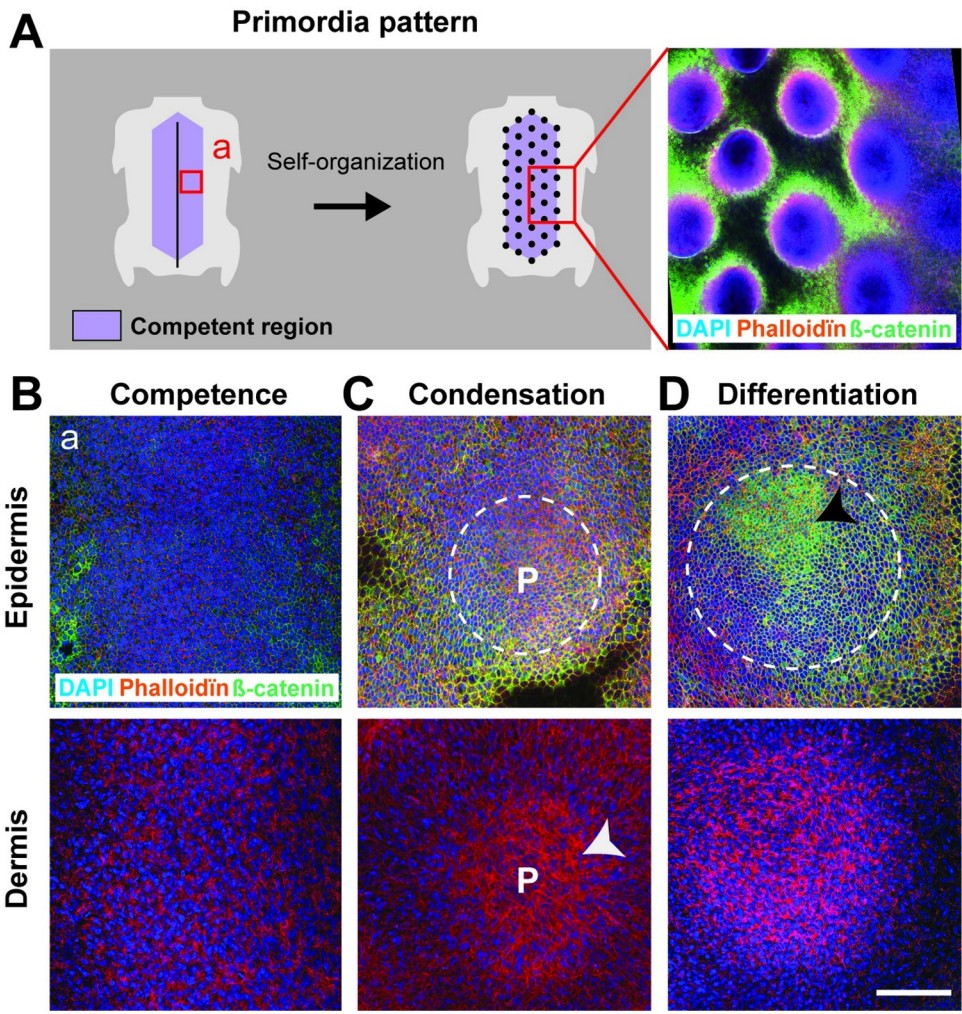

**Fig 1. Self-organized primordia patterning in the avian skin. (A)** Left panel: Competent regions (in purple) in birds' dorsal skin form primordia arrays (black dots) through cellular self-organization. Right panel: The primordia pattern was visible on a 10× confocal view (corresponding to the red square in the left panel) of a DAPI (in blue), β-catenin (in green), and phalloïdin (in red) stain in the Japanese quail embryonic skin. **(B–D)** The 40× confocal views at the position of the red square in A (noted a) and oriented antero-posteriorly show developing primordia. Prior to pattern formation (competence stage; B), epidermal and dermal cells are homogeneously distributed. Cells then compact locally (condensation stage; C), forming nascent primordia (P, white dotted lines in epidermal views, arrowheads in dermal views). Primordia initiate programs of feather production (differentiation stage; D) upon nuclear translocation of β-catenin in epidermal nuclei (which occurs initially in the anterior part of primordia; black arrow). Scale bars: 100 μm.

as readily round shapes marked by the spatial restriction of *ß-catenin* expression, forming a longitudinal dotted line, or primordia row. New rows were gradually juxtaposed laterally in a sequential wave, as previously described [23,26,33]. When ß-catenin translocated in epidermal nuclei within the first-formed primordia, marking the onset of their differentiation stage, at least 3 rows were visible, together producing the hexagonal motif typical of the feather follicle pattern of these species [23,26]. By contrast, in the emu *Dromaius novaehollandiae* and the ostrich *Struthio camelus*, both part of the ancestral-most group *Paleognathae*, initial competent regions were 2 wide *ß-catenin*-expressing areas on each side of the dorsal midline. The firsts primordia emerged visibly simultaneously in budding shapes distributed in wavy oblique lines. They progressively acquired rounder shapes, together producing a spatially random

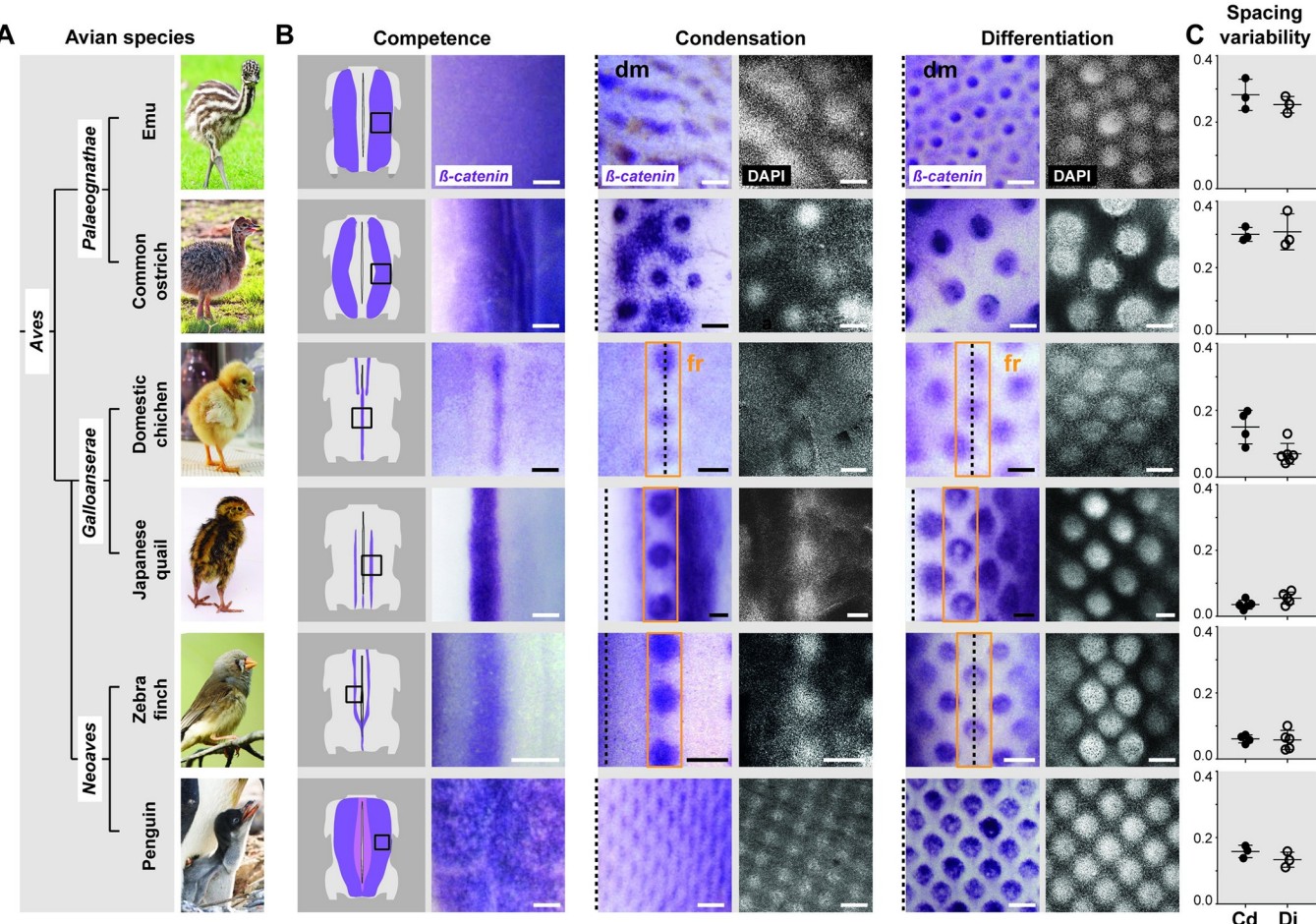

**Fig 2. Natural variation in primordia array geometry between avian species. (A)** Juveniles of chosen bird species represent the 3 major clades of the avian phylogeny. **(B)** The earliest distribution of *β-catenin* transcripts (in purple) in competent longitudinal segments or areas (i.e., at competence stage, first column) is shown on schematics of embryonic dorsum and images corresponding to black squares. At condensation stage (middle column), *β-catenin* was restricted to nascent primordia, also visualized with DAPI stains (in white, 10× confocal views), showing that species differ in timely dynamics of primordia emergence (i.e., in a row-by-row wave or simultaneously). At differentiation stage (right column), species-specific primordia array geometries had formed. Black dotted lines indicate the position of the dorsal midline (dm); orange boxes show the first-formed primordia row (fr) in species displaying a row-by-row wave. Scale bars: 200 μm. **(C)** Quantifying standard deviations of distances between adjacent primordia (spacing variability, in a dimensionless number defined in Materials and methods and see S2 Fig) along segments or areas at condensation (Cd) and differentiation (Di) stages showed that species vary in the regularity of their primordia pattern. Error bars: mean with standard deviation. The data underlying this figure can be found at 10.5281/zenodo.7006365. Photo credits: Jooin (emu), Barbara Fraatz (Pixabay, common ostrich), Publicdomain pictures (domestic chicken), Manceau laboratory (Japanese quail), Wikimedia (zebra finch), and Raphaël Sané (www.raphaelsane.com, Gentoo penguin).

primordia array. Finally, in the Gentoo penguin *Pygoscelis papua*, flightless *Neoaves* bird, the initial competent region was a medial *ß-catenin*-expressing area that rapidly formed a tight mesh of evenly distributed primordia (Figs 2B and S1). Thus, primordia geometries arise with varying dynamics (i.e., in a sequential wave or simultaneously, as in [26,33]) within species-specific patterning spaces (i.e., competent regions restricted to thin segments or wider areas). To quantify pattern fidelity, we recorded variability in spacing between adjacent primordia in species-specific patterning spaces outlined by initial *ß-catenin* expression (i.e., along first segments in the domestic chicken, Japanese quail, and zebra finch and across wider areas in the emu, ostrich, and penguin; see Materials and methods and S2 Fig). This method ensured we disentangled processes putatively involved in the progression of a wave to those controlling primordia pattern fidelity. We found that primordia emerged and remained regularly

distributed in initial segments/areas in the Japanese quail, zebra finch, and penguin, while they evolved from irregularly to regularly distributed in the domestic chicken, and displayed irregular distribution throughout time in the emu and the ostrich (Fig 2C). Thus, there is no correlation between pattern fidelity and the presence of a patterning wave. In addition, quantifications of primordia size and density showed that these attributes vary between species, but without correlation with differences in pattern fidelity (S3 Fig), suggesting that each pattern attribute is controlled independently. Together, these results evidenced inter-species variation in the timely evolution and outcomes of the self-organization of primordia in arrays.

## Inter-species differences in emergent cell anisotropy correlate with pattern variation

To identify tissue properties shaping pattern geometry differences, we compared the density and shape of epidermal and dermal cells prior to and during primordia formation. We found that local cell density is typical to each species, not linked to position in the phylogeny, and evolving through time without correlation with species-specific primordia size, spacing, or geometry (S4 and S5 Figs). This is consistent with previous work showing that cell density thresholds above which primordia emerge vary between birds [26,33]. To quantify cell shape, we marked skin tissues with phalloïdin and used a custom-made image quantification software based on Fourier transform (FT) without segmentation of cell contours [35]. It allowed us to quantify the anisotropy of average cell shape, defined here as the amplitude of deformation of skin cells and their angle with respect to the midline of the embryo (see Materials and methods). We found that in the epidermis of all species, cells were slightly elongated along the antero-posterior axis at competence stage (S6 Fig) and both within and between primordia at condensation (S7 Fig) and differentiation stages (S8 Fig). Their anisotropy amplitude values did not correlate with primordia pattern fidelity at any stage (S9 Fig). In the dermis however, we observed such correlation. Specifically, in the emu and the ostrich, species with low-fidelity patterns, dermal cells were isotropic at all stages (Figs 3A and 3B, and S10). In the domestic chicken, in which the primordia pattern progressively evolves from irregular at condensation stage to regular at differentiation stage, dermal cells in the competent segment (i.e., prior to primordia emergence) were first isotropic. At condensation stage, they became transiently anisotropic along the antero-posterior axis only inter-primordia spaces (remaining isotropic within nascent primordia), consistent with recent observations in this species [13]. Cell anisotropy then dropped at differentiation stage. Thus, in this species, cell anisotropy in the mesenchymal dermis, oriented along the axis of the competent segment, was transient and spatially restricted to inter-primordia regions (Figs 3C and S10). In the dermis of Japanese quail and zebra finch embryos, species in which the primordia pattern emerges and remains regular through time, we observed antero-posterior cell anisotropy as early as competent stage. At condensation stage, values of cell anisotropy amplitude dropped within nascent primordia and remained high in inter-primordia spaces. At differentiation stage, cells became isotropic in the inter-primordia space. Thus, in these species, dermal cells became anisotropic earlier than in the domestic chicken, correlating with the early high-fidelity of the primordia pattern (Fig 3D and 3E, and S10). Finally, in the penguin, dermal cells had strongest anisotropy values from competence stage to primordia differentiation. In addition, we observed 2 striking differences with other species also exhibiting dermal cell anisotropy: First, penguin dermal cells were elongated along the dorso-ventral axis, orthogonally to those of the other species. Second, cell anisotropy was maintained at differentiation stage in the penguin, contrary to other species in which cell anisotropy drops upon primordia differentiation. Thus, in the penguin, strong, non-transient dorso-ventral cell anisotropy correlated with extreme high-fidelity pattern (Figs

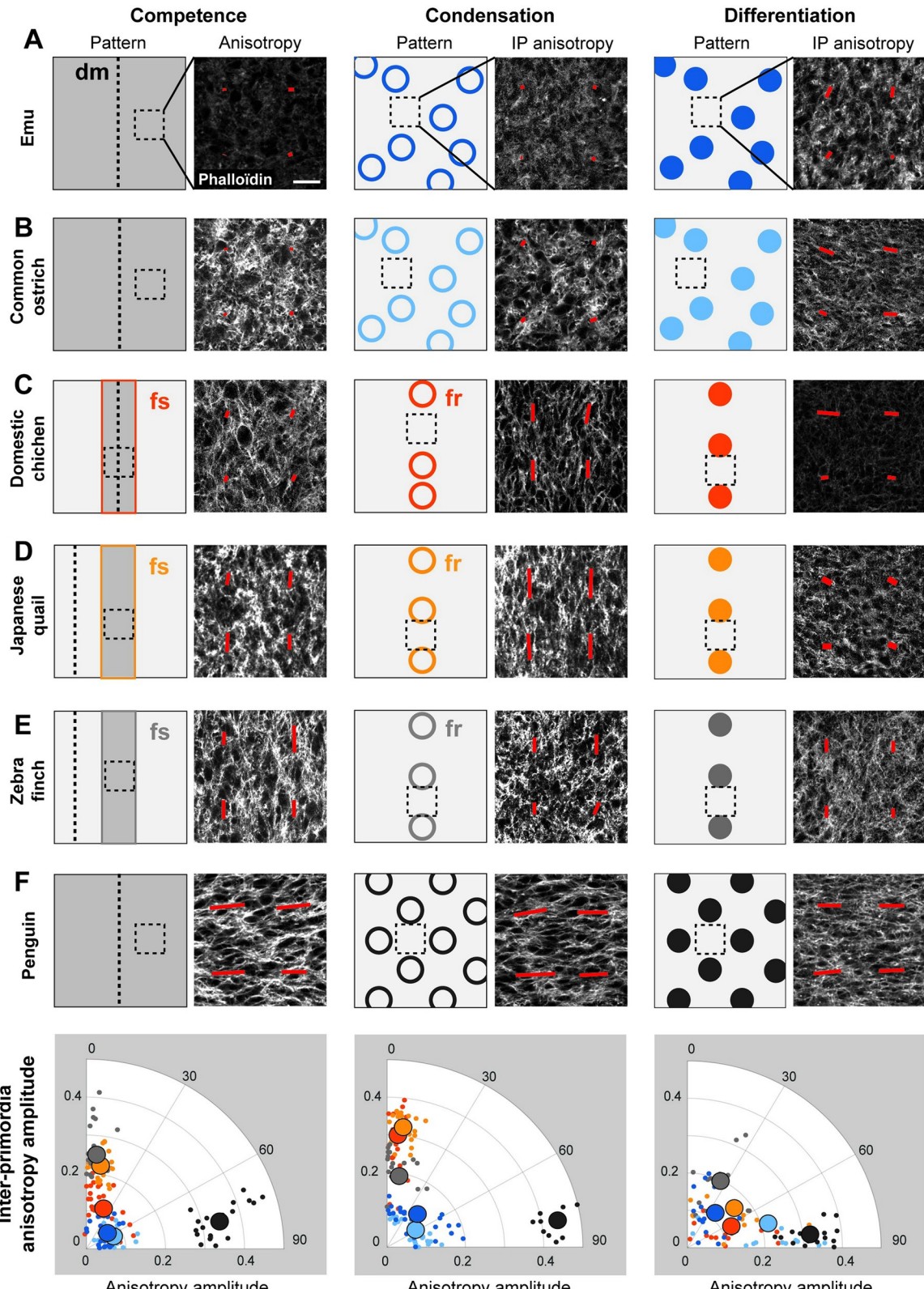

**Fig 3. Dermal cell anisotropy amplitude correlates with primordia array geometry. (A–F)** For each column (left, competence stage; middle, condensation stage; right, differentiation stage), left panels show schematics of the embryonic skin in emu (A), common ostrich (B), domestic chicken (C), Japanese quail (D), zebra finch (E), and penguin (F) embryos. Black dotted lines indicate the position of the

dorsal midline (dm); color-coded boxes or circles show the first segment (fs) and first-formed primordia row (fr) in species displaying a row-by-row wave. Right panels show confocal views at 40× of 100 μm² magnifications of phalloïdin-stained inter-primordia regions corresponding to black dotted squares in schematics in the embryonic dermis of each species. Scale bars: 20 μm. Primordia emergence involves dynamic changes in the anisotropy of average cell shapes, which we extracted using Fourier transform and represented by red bars ([35] and see Materials and methods; bar length is equal to measured anisotropy and bar direction is that of cell elongation). Bottom graphs show quantifications of anisotropy amplitude values (in a dimensionless number defined in methods) in all species, color-coded according to schemes and represented into polar coordinates for each stage (small dots are individual values, large dots are averaged values; $n$ = 3 specimen per species). The data underlying this figure can be found at 10.5281/zenodo.7006365.

3F and S10). These results demonstrated that the processes establishing cell anisotropy and controlling its directionality and timely evolution differ between species. Together, comparing values of dermal cell anisotropy amplitude between all species showed that they correlated with primordia spacing variability, but not with primordia size or density (S11 Fig). These observations suggested that emerging properties of cell anisotropy contribute to the self-organization of primordia such that they result in arrays with species-specific fidelity.

## Tissue shape changes do not impair self-organization but modify fidelity in its outcomes

To test this hypothesis, we studied pattern formation on cultured skin explants dissected at competence stage: In this ex vivo assay, substrate conditions are per se modified but the skin tissue develops autonomously and produces primordia patterns with proper timely dynamics [26]. We first quantified dermal cell anisotropy and final pattern fidelity in explants of Japanese quails, in which the primordia pattern forms within longitudinal segments (Fig 4A). We found that contrary to in vivo conditions in which the surface of the skin increases with age, cultured explants of Japanese quails slightly retract over time (i.e., approximately 10% decrease of their surface area; S12 Fig). Despite this difference, they produced primordia patterns with the same final geometry and spacing variability than in vivo (Fig 4B–4D). Primordia density was higher, likely due to explant retraction, and consistent with previous findings ([12] and S13 Fig). Japanese quail explants also recapitulated in vivo-like dynamics of cell shape anisotropy: antero-posterior cell anisotropy displayed high amplitude at condensation stage before dropping at differentiation stage (Fig 4E and 4F). In the emu, species in which the primordia pattern forms in a wide skin area, cultured explants maintained their initial size (S12 Fig). At differentiation stage, primordia density, size, and arrangement were undistinguishable from in vivo conditions (S13 and S14 Figs). Expectedly, dermal cells remained isotropic (S14 Fig). Thus, in Japanese both quails and emus, ex vivo conditions did not impair timely dynamics of cell shape anisotropy and the acquisition of proper pattern fidelity.

Culturing the skin ex vivo, however, affected self-organization's outcomes in the penguin. To study the latter, we used specimens of African penguins *Spheniscus demersus*, closely related to Gentoo penguins, and displaying identical dorso-ventral anisotropy and high-fidelity pattern (S15 Fig). Contrary to the Japanese quail and the emu, penguin explants visibly shrunk progressively, and at differentiation stage, their surface area had reduced by approximately 50% (S12 Fig). We quantified their pattern geometry at differentiation stage (Fig 4G). We found that primordia density increased and primordia size decreased, possibly due to the retraction of explants (S13 Fig). Strikingly, the high pattern fidelity observed in penguin skins in vivo was lost in explants (Fig 4H–4J). Penguin explant dermal cells were anisotropic, which is also the case at differentiation stage in this species in vivo, but the orientation of cell anisotropy strikingly varied depending on location. At places, cells were markedly oriented along the antero-posterior axis, in sharp contrast with in vivo conditions, penguin dermal cells normally displaying dorso-ventral orientation (see Fig 3F). Some areas of the explant however

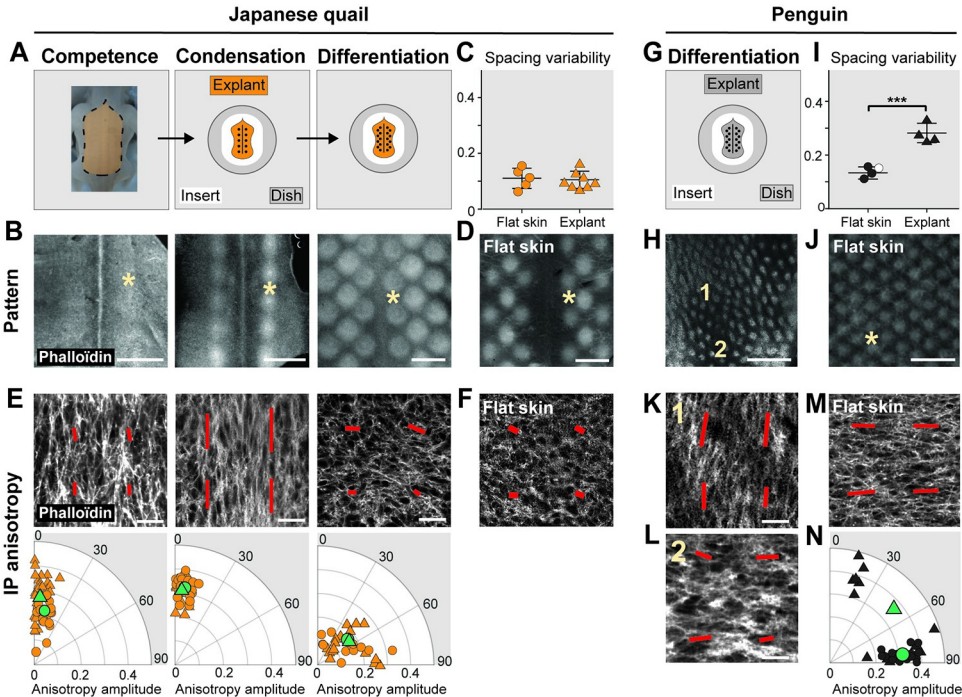

**Fig 4. Dermal cell anisotropy amplitude correlates with pattern differences between ex vivo and in vivo conditions. (A)** Dissected portions of dorsal embryonic skin (i.e., explant: shown here, a Japanese quail, orange region) at competence stage were placed on insert membranes (in white) in Petri dishes (in gray) and cultured to differentiation stage. **(B)** The 3,2× views of phalloïdin-stained cultured explants of Japanese quail embryos show they retain proper dynamics of primordia pattern emergence. **(C)** Quantifications of spacing variability show that at differentiation stage, pattern fidelity was maintained in explants (triangles; $n = 8$) compared to flat skins (dots; $n = 5$; unpaired 2-tailed $t$ test; $p = 0.7791$). **(D)** The 3,2× view of a phalloïdin-stained Japanese quail flat skin. **(E, F)** Confocal views corresponding to the location of asterisks in (B) and (D) and respective quantifications (as described in Fig 3) of phalloïdin-stained inter-primordia dermis show that in Japanese quail explants (E, triangles), cells remain anisotropic at competence and condensation stage and isotropic at differentiation stage, similar to control flat skins (F, dots). **(G)** African penguin explants were cultured to differentiation stage. **(H–J)** The 3,2× views of a phalloïdin-stained African penguin explant (H) and flat skin (J), and quantifications of spacing variability (I) show that in penguin explants, (triangles; $n = 4$), pattern fidelity was significantly impaired compared to control flat skins (dots; $n = 3$ for Gentoo penguins in black; $n = 1$ for African penguins in white, see Materials and methods; $p = 0.0005$). **(K–M)** The 40× confocal views of phalloidïn-stained explants showed that while cell shape anisotropy varied depending on location (2 positions are marked with (1) and (2) and shown in (K) and (L), respectively), it differed from that observed in flat skins (M). **(N)** Quantifications showed that the amplitude and orientation of cell shape anisotropy orientation were highly variable in explants as shown by large distribution of individual values (small triangles), global dorso-ventral anisotropy being lost (large triangle), compared to control flat skins (dots). Small data shapes are individual values, large data shapes (in green) are averaged values. The data underlying this figure can be found at 10.5281/zenodo.7006365. Error bars: mean with standard deviation. Scale bars: 500 μm (primordia pattern), 20 μm (anisotropy).

retained normal dorso-ventral anisotropy (Fig 4K–4N). Thus, in penguin explants, global dorso-ventral anisotropy was lost. These observations suggest that in the penguin, the maintenance of dorso-ventral cell anisotropy, likely linked to factors controlling overall tissue size such as contractile tissue properties, is required to produce a pattern with proper geometry. Together, comparative ex vivo experiments indicate that the process of primordia self-organization occurs regardless of differences in substrate conditions, overall tissue size, or the acquisition of cell shape anisotropy. However, this emerging cell property is associated to species-specific differences in patterning outcomes between cultured explants and in vivo conditions.

## Chemical perturbation of cell anisotropy modifies pattern fidelity

We took advantage of these differences to functionally assess the role of cell anisotropy. Japanese quail explants maintain normal temporal dynamics of cell anisotropy; we thus first impaired the acquisition of cell anisotropy prior to primordia emergence by treating skin explants with Latrunculin A, drug known to inhibit actin polymerization [36] (Fig 5A). We found that at low dose, drug treatment did not affect proper row-by-row pattern formation dynamics, confirming that cell anisotropy does not control patterning sequentiality. We measured pattern geometry at condensation and differentiation stages, and found that Latrunculin A treatment significantly increased spacing variability between primordia (Fig 5B). As expected, it did not modify local cell density but efficiently decreased dermal cell anisotropy amplitude (Figs 5C and S16). Thus, in the Japanese quail, impairing early cell shape anisotropy decreased pattern fidelity. Strikingly, when explants were treated with a 2-h pulse of Latrunculin A at competent stage rather than continuously throughout pattern formation, early dermal cell anisotropy was impaired but pattern fidelity was only transiently modified: it was lower at condensation stage but had entirely recovered at differentiation stage (S17 Fig). This is consistent with patterning dynamics observed in other species: In the domestic chicken, a delay in the acquisition of cell shape anisotropy is associated to the production of a transiently irregular pattern, its fidelity increasing only at differentiation stage. In the penguin, cell shape anisotropy is maintained at differentiation stage and its pattern fidelity is extremely high (see Fig 3).

As expected, Latrunculin A treatment did not significantly modify pattern fidelity in emu explants, dermal cells remaining isotropic similar to in vivo conditions (S14 Fig). In penguin explants, spacing variability was not further increased in drug-treated explants compared to control ex vivo conditions, though primordia displayed elongated shapes (Fig 5D and 5E). Dermal cell anisotropy was lower and less spatially variable than control explants (Fig 5F). Thus, drug-induced homogenization of low dermal cell anisotropy throughout the surface of penguin explants caused defects in primordia individualization. In all species, primordia had sizes and densities identical to control explants (S18 Fig). These results associated minimal amplitude value of initial dermal cell anisotropy with high fidelity of primordia arrangement.

## Mechanical perturbation of cell shape anisotropy modifies pattern fidelity

Second, because the loss of cell shape anisotropy in penguin explants is coupled to the shrinking of the skin (S12 Fig), we designed a mechanical test to externally influence cell elongation. To do so, we cultured competent dorsal skin portions on gels of collagen to which we applied controlled directional stretch prior to primordia emergence as in [37] (see Materials and methods). Approximately 8 h after applying to Japanese quail explants a stretch orthogonal to the normal axis of pattern formation (i.e., a dorso-ventral stretch; Fig 6A and 6B), anisotropy amplitude decreased compared to non-stretched control explants, and cells reoriented along the dorso-ventral axis (Fig 6C). Thus, applying external traction to developing explants efficiently perturbed the amplitude and orientation of dermal cell anisotropy. Dorso-ventrally stretched Japanese quail explants reached differentiation stage after 48 h (Fig 6D). Quantifications of pattern fidelity showed that variability in primordia spacing increased, consistent with the decrease in early anisotropy induced by stretch in this species (Fig 6E). At this stage in control explants, cells are isotropic; in stretched explants, they remained slightly oriented along the dorso-ventral axis (Fig 6F). These results confirmed those obtained in Latrunculin A-treated explants, further supporting that early, antero-posterior cell shape anisotropy is required for proper pattern fidelity along competent segments in Japanese quails. We used this experimental design on penguin skins to counteract the effect of explant shrinking observed in this species and re-induce the direction of normal anisotropy observed in vivo. Dorso-

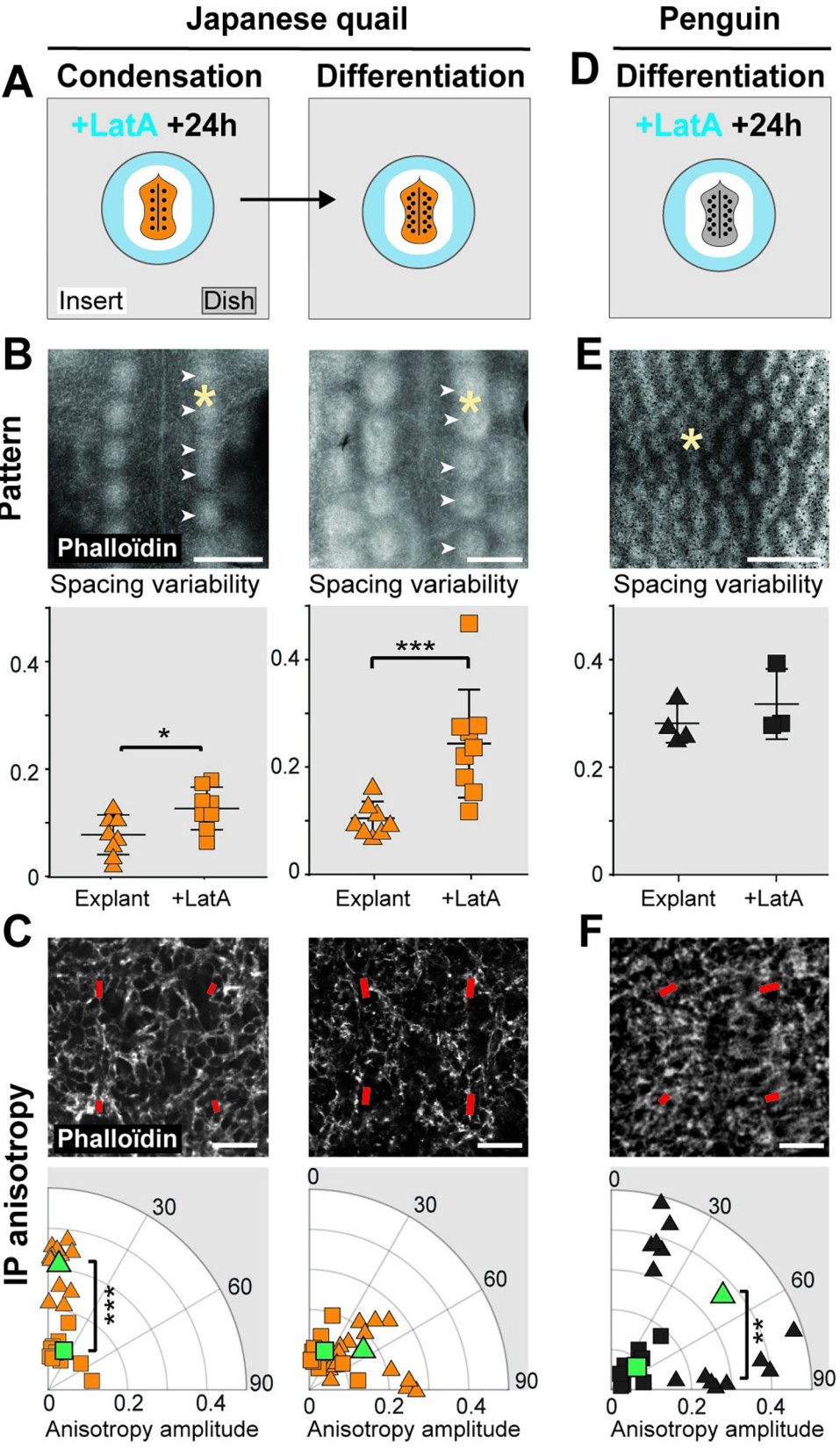

**Fig 5. Drug-induced decrease of cell anisotropy reduces pattern fidelity. (A)** Japanese quail skin explants prepared at competence stage (in orange) were treated with low doses of Latrunculin A (LatA, in blue) and cultured to condensation stage (+24 h; left panel) or differentiation stage (right panel). **(B)** Quantifications of primordia spacing variability on phalloïdin-stained control flat skins (explant; triangles) and LatA-treated cultured explants (+LatA; squares) of Japanese quail embryos showed that LatA treatment caused a significant decrease in pattern fidelity at condensation stage (explants, $n = 9$; +LatA, $n = 9$; unpaired 2-tailed $t$ test; $p = 0.0233$) and differentiation stage (explants, $n = 8$; +LatA, $n = 9$; $p = 0.002$). White arrowheads show emerging primordia. **(C)** At condensation stage in treated explants, antero-posterior anisotropy orientation was lost and anisotropy amplitude was significantly reduced (red bars, see Materials and methods and Fig 3; unpaired 2-tailed $t$ test; $p < 0.0001$ for amplitude and 0.67 for angle). At differentiation stage, cells remained isotropic similarly to untreated controls. **(D)** African penguin skin explants were treated with LatA as described in (A). **(E)** In drug-treated explants at differentiation stage, we observed a maintenance of the culture-induced low pattern fidelity (in black; explants, $n = 3$; +LatA, $n = 3$; $p = 0.3928$). **(F)** Dorso-ventral anisotropy was further impaired ($p = 0.0002$ for amplitude and 0.704 for angle). Asterisks show the location of 40× confocal views on explants. Small data shapes are individual values, large data shapes (in green) are averaged values. The data underlying this figure can be found at 10.5281/zenodo.7006365. Error bars: mean with standard deviation; significance of statistical tests for anisotropy amplitude is shown with stars. Scale bars: 500 μm (primordia pattern), 20 μm (anisotropy).

ventrally stretched penguin explants reached differentiation stage after 4 days (Fig 6G). We measured explant size and found it had recovered from culture-induced retractation, its surface area being comparable to corresponding portions of skins in vivo (S19 Fig). In addition, primordia spacing variability decreased to reach in vivo-like values (Fig 6H), and we observed a recovery of global dorso-ventral cell shape anisotropy (Fig 6I). Thus, applying an external stretch along the direction of normal anisotropy rescued pattern fidelity in penguins. Together, these experiments demonstrated that early cell shape anisotropy, acquired through mechanisms likely linked to mechanical stress on the developing skin tissue, is an initial condition of the un-patterned system, and indicated that it contributes to ensuring that self-organization results in high fidelity patterns.

## Early cell shape anisotropy and cell motility stabilize primordia positioning

To gain a better mechanistic understanding of the chain of events through which early dermal cell anisotropy ensures correct pattern geometry, we performed time-lapse imaging experiments in the developing skin of membrane-GFP (mb-GFP) Japanese quails (Fig 7A). We followed the behavior of dermal cells in the first-formed competent segments (i.e., prior to and during primordia emergence), as well as within un-patterned regions located laterally to competent regions. In competent segments, we synchronized movies by using as reference time point $t_{ref}$ the visible condensation of dermal cells forming nascent primordia and applied to movies custom-made image analysis software for anisotropy detection. We found that continuous mean values of cell anisotropy amplitude provided sufficient criterion to automatically detect the competent region, and within this space, the putative primordia or inter-primordia regions (S20 Fig). We analyzed the dynamic evolution of cell anisotropy in these software-defined areas. Consistent with results obtained on fixed tissues, we showed that in un-patterned regions, dermal cells are isotropic (Figs 7B and S21 and S1 Movie). At least 4 h before primordia emergence, in the presumptive region corresponding to the first segment, dermal cells became elongated along the antero-posterior axis, marking tissue competence. Dermal cell anisotropy was then maintained in inter-primordia spaces at $t_{ref}$ as local condensations became visible, dropping progressively in the presumptive primordia region (Fig 7C and S2 Movie). As expected, in movies of skin explants treated with low doses of Latrunculin A, we observed a decrease in cell anisotropy amplitude (Figs 7D and 7E, and S20 and S3 Movie).

We analyzed cell movement in all movies using image correlation analysis (i.e., particle image velocimetry; see Materials and methods). In un-patterned regions, cells randomly moved across the dermis with an average speed of 4.5 μm/h (Fig 7F). During primordia

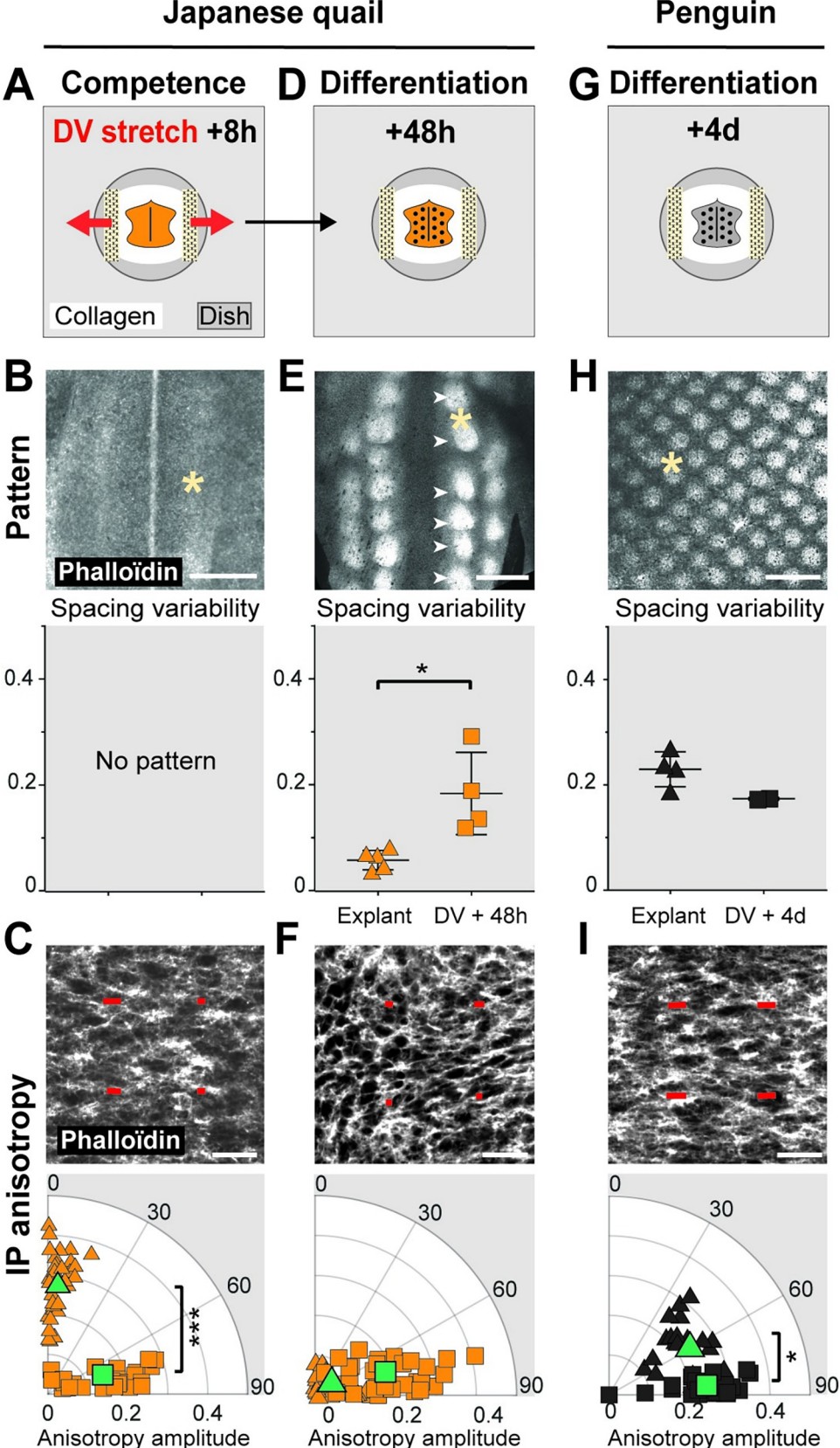

**Fig 6. Stretch-induced dermal cell anisotropy modifies explant pattern fidelity. (A)** Japanese quail skin explants at competence stage (in orange) were placed on collagen gels (in white) on Petri dishes (in gray) and stretched along the dorso-ventral axis by pulling on Velcro bands (in yellow) attached to the gels as in [37] (and see Materials and methods). They were cultured during 8 h (to assess cell shape anisotropy). **(B)** No pattern was visible in stretched explants after 8 h. **(C)** Compared to control explants (triangles; $n = 4$), dorso-ventral stretching significantly reduced antero-posterior cell shape anisotropy after 8 h in Japanese quail explants (squares; $n = 12$, unpaired 2-tailed $t$ test; $p < 0.0001$ for both anisotropy amplitude and angle). **(D)** Stretched Japanese quail explants were observed after 48 h (i.e., at differentiation stage). **(E)** Primordia spacing variability was significantly reduced in explants 48 h after stretching (squares; $n = 4$) compared to control explants (triangles; $n = 5$). White arrowheads show emerging primordia. **(F)** Cell shape anisotropy was strikingly oriented along the dorso-ventral axis (i.e., orthogonally to dermal cells in un-stretched explants, $p < 0.0001$ for both anisotropy amplitude and angle). **(G)** African penguin skin explants were stretched and studied after 4 days (i.e., at differentiation stage). **(H)** Spacing variability displayed control-like values (explant: $n = 2$; DV+4h: $n = 4$, $p = 0.043$). **(I)** Global dorso-ventral orientation of cell anisotropy was restored ($p = 0.043$ for anisotropy angle, $p = 0.862$ for anisotropy amplitude). Asterisks show the location of 40× confocal views on explants. Small data shapes are individual values, large data shapes (in green) are averaged values. The data underlying this figure can be found at 10.5281/zenodo.7006365. Error bars: mean with standard deviation; significance of statistical tests for anisotropy amplitude is shown with stars. Scale bars: 500 μm (primordia pattern), 20 μm (anisotropy).

individualization in the first competent segments, cell velocity increased to an average speed of 10 μm/h (Fig 7G and 7H), consistent with values obtained in previous avian studies [33] and data obtained in rodents [38]. Cell velocity was however stable throughout the patterning space (i.e., we did not detect differences between presumptive primordia and inter-primordia regions). Thus, an increase in cell motility is concomitant to the acquisition of patterning competence and dermal cell shape anisotropy. Strikingly, when skin explants were treated with Latrunculin A, we observed a significant increase of dermal cell motility across the whole patterning space, compared to un-treated explants (Fig 7G and 7H). These results contrasted with effects obtained with higher doses of Latrunculin A, known to block cell movement [36,39]: Here, cells likely release interaction abilities due to actin perturbation, which results in freer displacement. Thus, the acquisition of cell shape anisotropy limits the increase in cell motility marking the onset of primordia morphogenesis within the competent segment. We performed divergence analyses in all movies and showed that Latrunculin A-treated explants undergo lower compaction in the forming primordia region compared to control explants (S20 Fig). This suggests that drug treatment induced cell condensation defects.

We followed the dynamic position of nascent primordia along the antero-posterior axis in the first segment using a custom automatic detection and tracking method (Figs 7I and S20 and see Materials and methods). We found that in control explants, primordia emerged at stable and regularly spaced positions through time (Fig 7J and S2 Movie). However, in movies of skin explants treated with low doses of Latrunculin A, primordia displayed higher overall displacement. They individualized from transient elliptic shapes or small intercalary structures that sometimes fused together, in dynamics resembling those observed in the emu and the ostrich, before stabilizing in a pattern with low fidelity (Fig 7K and S3 Movie). Thus, a reduction of cell shape anisotropy increases primordia displacement and impairs processes of primordia individualization. Together, time-lapse experiments show that cell shape anisotropy and cell motility are initial properties of the competent skin tissue that modulate primordia displacement and processes of primordia individualization.

## Discussion

This study uncovers cellular dynamics that control the fidelity of self-organization in a developing tissue. Here, early cell shape anisotropy modulates the behavior of mesenchymal cells in skin regions competent to undergo primordia morphogenesis, locally controlling the positioning of cellular aggregates marking the onset of primordia formation. Higher values of cell

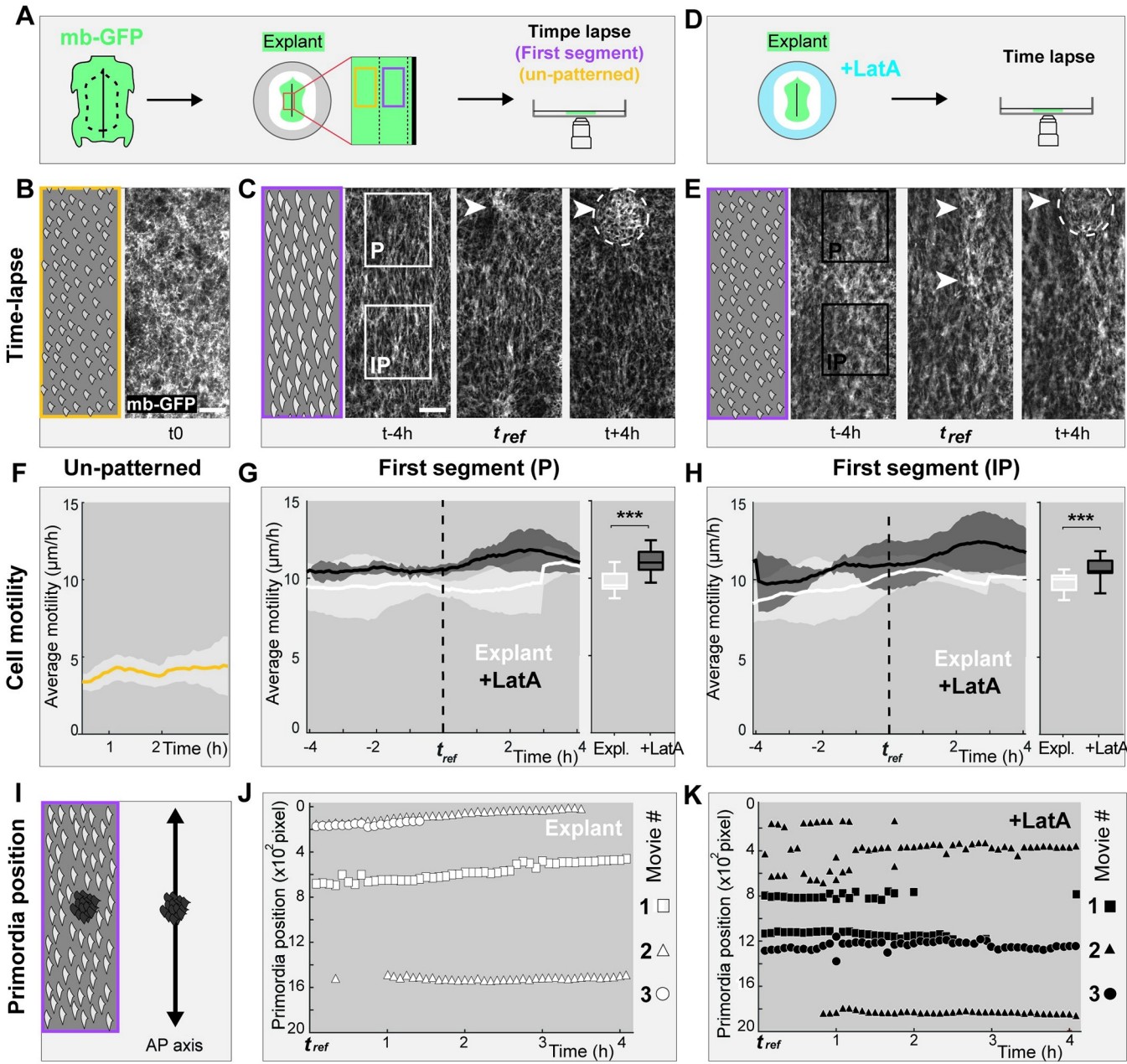

**Fig 7. Cell shape anisotropy and motility stabilize nascent primordia positioning. (A)** To perform time-lapse experiments, dorsal skin explants of membrane-GFP (mb-GFP) Japanese quails were placed dermal side down on nitrocellulose filters for confocal imaging (see Materials and methods). Movies focused on un-patterned regions (in yellow) or the first competent segment (in purple). **(B)** Snapshot at t0 of a time-lapse confocal movie in the un-patterned region and corresponding schematic. **(C)** Snapshots of a confocal movie in the first-formed segment 4 h before, at, and after $t_{ref}$ recorded at dermal level show the formation of a primordium (white arrow). A schematic shows anisotropic cells at t-4h. White squared lines mark the position of automatically detected putative primordium (P) and inter-primordium (IP) areas. White dotted lines and arrowheads show a nascent primordium. **(D, E)** The same experiments were performed after Latrunculin A treatment (LatA; black squared lines mark P and IP areas; white dotted lines and arrowheads show 2 nascent primordia). Scale bar: 50 μm. **(F)** Quantifications of average cell motility (in μm/h) during 2 h in the un-patterned region showed it is stable through time. **(G, H)** Quantifications of average cell motility (in μm/h) in hours to $t_{ref}$, marked with a line or averaged in box plots in IP and P regions showed that cell motility increases upon drug treatment (in black) compared to control conditions (in white; unpaired 2-tailed $t$ test, $p < 0.0001$ both for IP and P). **(I, J)** We tracked the antero-posterior position (in pixel ×10²) of software-detected primordia within competent segments. Graphical representations of primordia position at each time point of 3 independent movies (shown by a white shape code) showed that in un-treated explants, emerging primordia maintained stable positions. In movie #2 (triangles), 2 spaced-out primordia emerge. **(K)** In Latrunculin A-treated explants, the reduction of cell shape anisotropy caused imprecision in nascent primordia position, which can emerge closely and fuse together. This is seen in movie #2 (triangles), and #3 (dots). The data underlying this figure can be found at 10.5281/zenodo.7006365. Scale bars: 50 μm. Error bars: standard deviation, stars show significance of statistical tests.

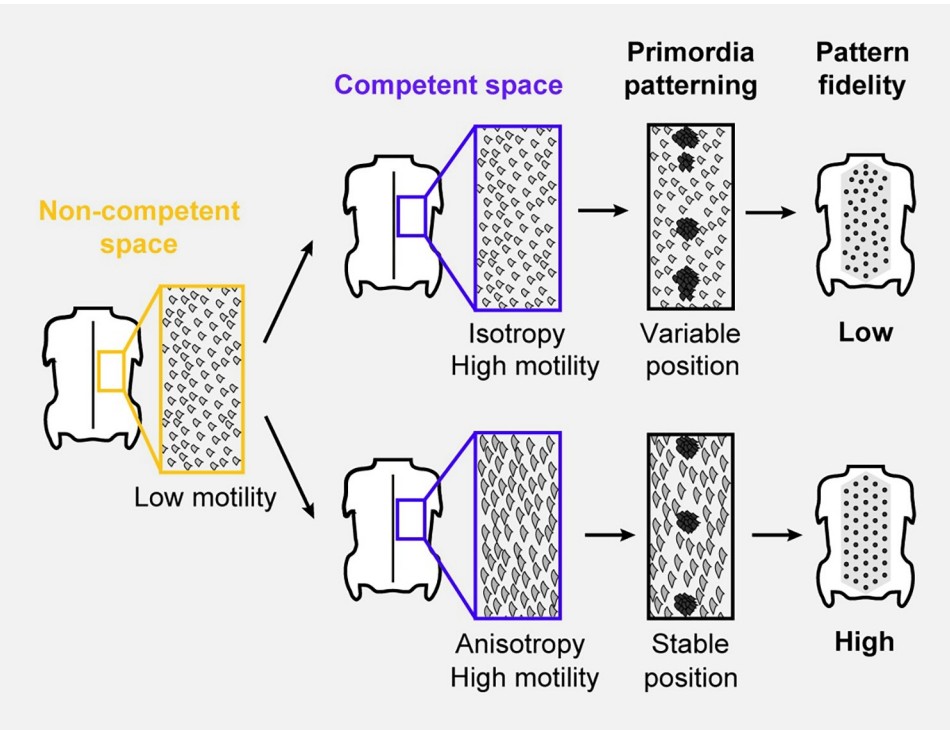

**Fig 8. Natural variation in cell shape anisotropy causes differences in pattern fidelity.** We propose a mechanism of pattern fidelity establishment in which non-competent patterning spaces (in yellow), characterized by low cell motility and isotropic cells, can undergo 2 scenarios, depending on species. In emus and ostriches, competence spaces (in purple) remain isotropic, and primordia position is variable, which results in low-fidelity patterns. In Japanese quails, domestic chicken, zebra finches, and penguins, competence spaces acquire early cell shape anisotropy, which constrains motility in dermal cells and stabilizes the position of nascent primordia, resulting in high-fidelity patterns.

anisotropy, such as observed in Japanese quails, zebra finches, or penguin embryos, stabilize primordia positioning and sharpen their individualization, which results in high-fidelity patterns. An absence of cell anisotropy, such as observed in the emu and the ostrich, or in drug-treated skin explants, leads to more variable dynamics of primordia positioning, which results in low-fidelity patterns (Fig 8).

We found that cell shape anisotropy is initially homogeneous across the competent region. In addition, recent work in the domestic chicken showed that cell shape anisotropy is present in dissociated dermal cells that retain the ability to self-organize into aggregates in vitro [13]. Together, these observations indicate that cell shape anisotropy represents an intrinsic and global property of competent skin regions. Strikingly however, this property is not required for pattern formation: In emus and ostriches, mesenchymal cells are isotropic but primordia arrays spontaneously emerge in the developing skin. Moreover, in Japanese quail explants, drug-induced isotropic mesenchymes produce primordia patterns with timely dynamics that do not markedly differ from unperturbed conditions. The robustness of primordia pattern emergence, typical of a self-organized system [1–5], is likely ensured by a combination of genetically programmed molecular signaling and tissue mechanics [14,22]. This aspect has complicated the identification of events controlling primordia pattern fidelity: Experimental perturbations through the *mis*-expression of candidate factors or the inhibition of cell behavior rarely entirely prevent pattern emergence, rather resulting in a spatial disorganization of primordia arrays that can be interpreted as a modification of processes controlling pattern fidelity

but that may instead result from changes in the morphogenesis and timely dynamics of self-organization itself, or vice versa. Here, we combined ex vivo functional assays with comparative analyses of natural variation, a correlative framework that allowed distinguishing self-organization per se from events shaping pattern fidelity. Though drug treatment or stretching may cause several cellular changes, our approach allowed us to link early cell shape anisotropy to final pattern fidelity.

Previous theoretical work predicted that the developing skin tissue become responsive to activator and inhibitor signals causing self-organization when it reaches a "competence" threshold established by species-specific cellular properties of the environment [26,33,40]. A parameter describing competence can numerically influence the fidelity of dotted patterns [26], and results of our study suggest that it may be represented by cell shape anisotropy. By surveying different species, we found that the timing of pattern stabilization correlates with the stage at which dermal cells become anisotropic. In the Japanese quail for example, in which the pattern is readily regular, cell shape anisotropy is acquired hours before the first mesenchymal aggregates become visible. Despite not being required for the tissue to become competent, global cell shape anisotropy may thus accelerate or facilitate the ability of the tissue to undergo self-organization. The local geometry of feather primordia has been previously shown to depend on the presence or absence of a global patterning event [40]; here, we hypothesize that pattern fidelity is not directly dictated by parameters of self-organization, but rather by the presence or absence of cell shape anisotropy that governs the onset and temporal dynamics of self-organization. The timely acquisition of the competence threshold has previously been suggested to depend on the presence or absence of a wave of molecular signaling [33,40]. An appealing hypothesis is that the signaling wave modulates the timely acquisition of cell shape anisotropy, itself controlling the competence threshold. This is consistent with results showing that dissociated mesenchymal cells retain cell shape anisotropy [13]; when they are cultured to produce embryonic chicken skins, primordia form simultaneously [41]. Future work taking advantage of differences between species displaying a temporal wave of segment-by-segment patterning will help understand how cell shape anisotropy may optimize the timely acquisition of competence.

We observed that the natural gradient in cell shape anisotropy amplitude was linked to both global pattern regularity and local primordia individualization: The stronger the initial amplitude of cell shape anisotropy, the more stable the overall primordia positioning and the sharper the local definition of cellular aggregates into individual primordia. It is therefore likely that cell shape anisotropy imposes amplitude-dependent constraints throughout competent regions that convert local self-organizing events responsible for primordia emergence into large-scale order.

Inter-species comparison and time-lapse experiments indicated how such global cell behavior may be transposed to spatially precise local morphogenetic events. First, we found that the antero-posterior amplitude of cell shape anisotropy in the Japanese quail is intricately linked to cell motility dynamics: Both cell anisotropy and motility increase in initial competent segments compared to naïve regions, and cell motility further increases when cell anisotropy is impaired by drug treatment. This strongly suggests that cell shape anisotropy controls pattern fidelity by limiting cell motility prior to and during self-organization. Previous experimental and theoretical evidence indeed showed that the initial aggregation of dermal cells into primordia is motility dependent (i.e., controlled by ERK-activity-driven chemotaxis [29]). Cell shape anisotropy can modulate cell motility through changes in the actomyosin network: anisotropic cells expose cryptic-binding sites for cytoskeletal proteins involved in cell-cell or cell-ECM adhesion complexes [42], which connects isolated cells and influences the production and composition of the ECM [43,44]. This modifies elastic and viscous properties of the

tissue [45], which results in differences in tissue stiffness that have a direct impact on cell movement and diffusion properties of the tissue [46]. It is appealing to apply this scenario to the developing avian skin because substrate stiffness has been previously shown to control primordia spacing in domestic chicken skin explants [12]. We can thus hypothesize that in bird species with strong cell anisotropy throughout competent patterning spaces, the dermal mesenchyme is globally stiff, which locally limits cell movement, and thus the spatial range at which dermal cells initially aggregate into primordia. Species with low cell anisotropy produce soft mesenchymes, allowing freer cell movement, which increases the displacement of initial cellular aggregates. Second, we found that cell shape anisotropy modulates the shape of primordia and the timely dynamics of their individualization: In emus, the absence of cell shape anisotropy is associated to an indistinct and gradual emergence of primordia through transient elliptic structures that progressively separate, and in Japanese quail explants, a reduction of cell shape anisotropy lowers local compaction, while in penguin explants, drug-induced homogenization of low dermal cell anisotropy throughout the competent skin surface causes local defects in primordia individualization. It is thus possible that cell shape anisotropy confers given contractile properties such that cell aggregates rapidly emerge from a mesenchymal continuum. This hypothesis is in line with data showing that in vitro, calcium-dependent cell contractility is required for proper aggregation in primordia-like structures [13]. Finally, cell shape anisotropy may also facilitate the generation, integration, and propagation of chemical cues on which the self-organized system relies, such as molecules of the Wnt, Bmp, and FGF families [7–10,29,30,40,41]. To better understand how cell shape anisotropy and other emergent cell properties such as cell motility coordinate the cellular and tissue response to molecular signaling and tissue mechanics, future experimental approaches benefiting from natural variation and in-depth live imaging shall associate species-specific values of cellular dynamics to inter-species differences in the dynamic expression of candidate morphogens, material properties of the self-organizing patterning space, and local values of mechanical stresses.

Such experiments will also be necessary to identify events acting upstream of cell shape anisotropy. These mechanisms may involve intrinsic contractile properties of mesenchymal cells, recently demonstrated in vitro to align the ECM, which in turns causes cell elongation [13]. Similarly, cell shape anisotropy may be produced by externally or intrinsically generated anisotropic tensile forces in the developing skin, consistent with the effect of externally mediated traction on cell shape changes in skin explants, with our unique observations in penguins, which display dermal cell shape anisotropy oriented along the dorso-ventral axis that is lost in skin explants, and with previous studies involving these forces in pattern geometry [47–49]. Cell shape anisotropy may also be linked to the shape of the patterning space, consistent with our previous theoretical predictions [26]. In species in which patterning spaces are thin longitudinal segments, antero-posterior cell elongation may result from steric constrains or be driven by upstream instructive signals creating a gradient of positional information along the antero-posterior axis.

Finally, from an evolutionary perspective, our work shows that collective cellular properties are key to ecologically relevant canalization of self-organization. Our comparative approach allows suggesting they shape feather pattern geometry evolution. In low pattern fidelity flightless *Paleognathae*, delayed competence acquisition in response to molecular signaling [34] may be due to the absence of cell anisotropy and free cell movement. In the alien case of penguins, extreme anisotropy allows achieving highest feather pattern regularity, key to water resistance, in the absence of spatial restriction of the patterning space. Cell shape changes may constitute an evolutionary novelty in this *Neoaves* bird adapted to aquatic life and extreme weather conditions. Further exploring mechanisms of pattern formation in these species will continue shedding light on the developmental basis of adaptation in birds.

## Materials and methods

### Embryo sampling

Japanese quail, domestic chicken, emu, and common ostrich fertilized eggs were obtained from local suppliers, respectively, Cailles de Chanteloup, Les Bruyères Elevage, Emeu d'Uriage, and Autruche de Laurette/Autruche du Père Louis/Autruche de la Saudraye. Zebra finch fertilized eggs were collected from a breeding colony at the Collège de France. Gentoo penguin eggs were harvested from natural breeding sites of Stevely Bay, Grave Cove and Weddell Island in the Western Falkland Islands. This allowed obtaining a large number of Gentoo penguin eggs specimens, and this species was thus used for descriptive analyses of pattern geometries, cell density, and cell anisotropy. African penguin eggs were provided by the La Palmyre Zoo. These specimens were available in low numbers but allowed controlled incubation and laboratory work; this species was thus used in explant assays. All animal work was performed in compliance with regulations for animal use and specimen collection of the French Government and the European Council. The welfare of the zebra finch breeding colony was guaranteed through regular care and visits approved by official and institutional agreement (Direction Départementale de la protection des populations and Collège de France, agreement C-75-05-12). Research licenses for Gentoo penguin specimen collection have been granted by the Environmental Planning Department of the Falkland Islands Government (#R26.2017; #R43.2018; #R36.2019).

### Flat skins and cultured explants preparation

Flat skins specimens were prepared as described [50] after egg incubation in Brinsea OVA-Easy 190 incubators in the laboratory, embryo dissection, and fixation in 4% formaldehyde. Skin explants were produced by dissecting dorsal portions of embryonic skins at competence stage, namely HH32 in the emu, HH29 in the Japanese quail, and HH30 in the African penguin (see justification for using this species in the first paragraph). We quantified the skin explant sizes at competence stage (i.e., upon preparation) and at differentiation stage by outlining their surface manually using the Fiji software polygon selection tool and recording the obtained area value (control explants: Japanese quail, $n = 6$; emu, $n = 3$; penguin, $n = 4$; stretched penguin explants, $n = 3$). For drug treatments, skin explants were placed dermal side down on culture insert membranes (Falcon #353103, #08-771-20) in Petri dishes over 800 to 1,600 µL DMEM solution supplemented with 2% FCS and 2% Penicillin/Streptomycin. Approximately 0.0625µM/ml Latrunculin A (Sigma #428021) was added to the culture media in a 2-h pulse or continuously until differentiation stage. Japanese quail explants were cultured on insert membranes at 37˚C in 5% $CO_2$ atmosphere (Thermo Scientific Midi 40) during 24 h (i.e., to condensation stage; $n = 8$ for control explants; $n = 9$ for continuously treated explants; $n = 6$ for pulse-treated explants) or 48 h (i.e., to differentiation stage; $n = 9$ for both control and continuously treated explants; $n = 3$ for pulse-treated explants). African penguin and emu explants were cultured on insert membranes during 96 h (i.e., to differentiation stage; $n = 6$ and 4, respectively, for both control and drug-treated explants). For stretching experiments, skin explants were cultured on 80% collagen gels (Corning) previously poured on Velcro bands as in [37]. Stretch was exerted manually by pulling on Velcro bands parallelly or orthogonally to the axis of the explant and fixing Velcro bands to the Petri dish using dissection pins (see Fig 5A). Japanese quail explants were cultured for 8 h to assess cell shape anisotropy ($n = 4$ control; $n = 12$ stretched explants). Japanese quail and African penguin explants were cultured to differentiation stage (during 48 h and 4 days, respectively) and assessed for pattern fidelity ($n = 5$ and $n = 4$ controls; $n = 4$ and $n = 2$ stretched explants).

## Imaging

Fixed flat skins and whole embryos were imaged using an AF-S Micro NIKKOR 60-mm f/2.8G ED macro-lens equipped with a D5300 camera (Nikon) and an MZ FLIII stereomicroscope (Leica) equipped with a DFC 450C camera (Leica). Confocal images were obtained using an inverted SP5 microscope (Leica) with 10× (dry) or 40× (immersed oil) objectives. For time-lapse imaging, dissected dorsal skin regions from membrane-GFP Japanese quails produced at the Pasteur Institute (Dr. Gros laboratory) at HH29, which corresponds to competence stage Co, were placed dermal side down on Millipore-nitrocellulose filters (Sigma #HABP04700) and cultured in a Lumox dish (ø50 mm, Sarstedt #11008) containing 5 ml of DMEM supplemented with 2% FCS and 2% Penicillin/Streptomycin (and when applicable, 0.0625 μM/ml Latrunculin A) at 37˚C in 5% $CO_2$ atmosphere. Time-lapse images were acquired every 5 min up to 10 h with a SPINNING DISK-W1 (Zeiss) equipped with a 25× (immersed oil) objective.

## Quantification of pattern attributes

To assess primordia density at condensation and differentiation stages in the emu ($n$ = 5 and 3, respectively), common ostrich ($n$ = 4 and 3), and Gentoo penguin ($n$ = 3 and 3, see justification for using this species in the first paragraph), species in which primordia form an array, we quantified the number of primordia per 1 $mm^2$ squares. For the domestic chicken ($n$ = 4 and 6), Japanese quail ($n$ = 8 and 8), and zebra finch ($n$ = 5 and 4), species in which primordia arise in longitudinal lines, we quantified the number of primordia in 0.3 mm × 1 mm rectangles within the first formed row of feather primordia and normalized values per $mm^2$. We quantified primordia size at condensation and differentiation stages by manually outlining DAPI-stained dermal condensations with the freehand selection tool in Fiji software and recording obtained area value with Fiji measurement tools. The 3 primordia of the first row or spread out in the array were measured for 3 different embryos in each species. Regularity of primordia arrangement was quantified at condensation and differentiation stages in the domestic chicken ($n$ = 4 and 6, respectively), Japanese quail ($n$ = 5 and 5), and zebra finch ($n$ = 5 and 5) as the standard deviation of distances (in μm) between neighboring primordium centers in the first formed row, recorded on imaged specimens using Fiji measurement software and normalized to the mean value of spacing (in μm). In the emu ($n$ = 3 and 3), common ostrich ($n$ = 3 and 3), and Gentoo penguin ($n$ = 3 and 3), primordia centers were detected automatically using a custom Matlab program based on image threshold adjustments (Dotfinder; [26]). Rare detection errors were manually corrected and the MATLAB function "delaunay" was applied to the resulting set of points. The Delaunay triangulation was used to extract the distance between neighboring primordia constituting a triangulation edge (in pixel); for each skin, we quantified pattern regularity by computing the standard deviation of these edge lengths (edges with a vertex positioned on the boundary were excluded from the analysis) normalized to their average (in pixel). This quantification method is illustrated in S2 Fig.

## Expression analyses

In situ hybridization experiments were performed as described previously [51] using antisense riboprobes synthesized from vectors containing 881-bp, 501-bp, and 685-bp fragments of coding sequences for *β-catenin* of the Japanese quail (also used for detection of expression domestic chicken, emu, and common ostrich), zebra finch, and Gentoo penguin, respectively. Digoxigenin-labeled riboprobes were revealed with an anti-digoxigenin-AP antibody (1:2,000, Roche) and an NBT/BCIP (Promega) substrate. Sequences of *β-catenin* primers were: F, AGCTGACTTGATGGAGTTGGA and R, TCGTGATGGCCAAGAATTTC for the Japanese

quail, F: TAGTTCAGCTTTTAGGCTCAGATG and R: CCTCGACAATTTCTTCCATACG for the zebra finch, and F, GAACATGGCAACCCAAGCTG and R, GCCTTCACGGT-GATGTGAGA for the Gentoo penguin.

## Immunohistological stains

Embryonic flat skin specimens were fixed in 4% formaldehyde overnight at 4˚C, rinsed, and stained using anti-β-catenin (Abcam ab2365; 1:100) and goat anti-rabbit Alexa 488 (Abcam 11008; 1:500) antibodies. Co-stains were performed using phalloïdin coupled to Alexa 546 (Abcam A22283; 7:200) and DAPI (Southern Biotech). Flat skins were mounted in Fluoromount (Southern Biotech) on slides prior to imaging.

## Quantification of cell density and anisotropy

To quantify cell density, we counted the number DAPI-stained cell nuclei within 3 squares of 100 $\mu m^2$ distributed along the antero-posterior axis on focal plans of confocal images at epidermal or dermal levels ($n$ = 3 per species). To extract coarse-grained cell anisotropy, we used our published FT-based program (available on GitHub [35]) on images segmented in 25 × 25 μm interrogation boxes with 50% overlap, as follows: We applied to each box a multiplying function to avoid singularities in the FT due to boundaries and computed the FT using the MATLAB fast FT algorithm (fft2.m). Resulting Fourier space patterns were binarized by retaining only the 5% brightest pixels yielding filled ellipses with correct aspect ratio, and inertia matrices were computed and diagonalized. This allowed extracting pattern anisotropy defined by $\frac{1}{2} ln \frac{L_{maj}}{L_{min}}$ (where $L_{maj}$ and $L_{min}$ are respectively major and minor axes of the ellipse), a dimensionless number equal to zero when the pattern is isotropic and whose value corresponding to average elongation of cells within a given interrogation box is represented by the length and orientation of a bar (shown in red in figures; see [35] for further details). Primordia and inter-primordia regions were manually defined as groups of 4 interrogation boxes using stains mentioned above. Anisotropy amplitude was plotted as individual anisotropy bars for each group of 4 interrogation boxes (small shapes in polar plots); averaged anisotropy amplitude values were plotted as the mean of all bar lengths for a given species (large shapes in polar plots and correlation plots).

## Quantification of time-lapse imaging experiment

For time-lapse analyses, evolving values of anisotropy allowed defining regions automatically. Briefly, we computed average anisotropy for each interrogation box of each image throughout movies in which drift in "x" and "y" axes had been corrected using the "Stackreg" plugin in Fiji software. We defined the competent segment as the 3 adjacent columns with maximal averaged anisotropy. Within the herein defined competent segment, putative inter-primordia regions were the 4 interrogation boxes with maximal average anisotropy and primordia regions those with minimal average anisotropy. Anisotropy amplitude values in each region were first plotted separately for all movies. Resulting curves were smoothed using a moving average over 30 frames (i.e., 2 h 30 mins) and interpolated in the primordia region using MATLAB function "interp1" with evenly spaced query points. Movies of control and LatA-treated explants were compiled using as reference time point $t_{ref}$ the first visible condensation of cells in 1 primordium, also allowing computing and plotting dynamics of average anisotropy amplitude. For analyses of cell movement, we used MATLAB particle image velocimetry analysis program MatPIV on movies we treated with a 2-pixel radius median filter of Fiji software to eliminate aberrant vectors. We used reference time points $t_{ref}$ and averaging methods

as above, primordia and inter-primordia regions were enlarged to 9 interrogations boxes (using the third column of the competent segment along the dorso-ventral axis and towards the other regions along the antero-posterior axis). The vector field obtained from MatPIV was used to quantify cell motility (averaging the norms of the 9 vectors in each region) and to assess divergent/convergent behaviors (using MATLAB function "divergence" and averaging over the 9 vectors in each region). Tracking of primordia along the antero-posterior axis was carried out using the same threshold-based method as before (Dotfinder; [26]): For each image, we retained the 30% brightest pixels, applied Gaussian filtering (using Matlab function "imfilter"), binarized the image, and detected primordia as white areas comprising at least 10,000 pixels. All these methods are illustrated in S20 Fig.

## Supporting information

**S1 Fig. Dynamics of primordia emergence in studied species.** The 40× confocal views located within competent skin regions of emu, common ostrich, domestic chicken, zebra finch, and penguin flat skins and oriented along the antero-posterior axis, show DAPI (in blue), β–catenin (in green), and phalloïdin (in red) stains. Similarly to the Japanese quail (see Fig 1), epidermal and dermal cells compacted locally in primordia (P, white dotted lines in the epidermis, arrowhead in the dermis) at condensation stage and initiated programs of feather production upon nuclear translocation of β-catenin in epidermal nuclei (white arrows) at differentiation stage. Scale bars: 100 μm.
(JPG)

**S2 Fig. Quantification of spacing variability in the emu, common ostrich, and penguin.** The 10× confocal views of DAPI stains (in white) in flat embryonic skins of emu, common ostrich, and penguin at differentiation stage and associated positions of feather primordia centers (black dots) detected by applying a custom-made MATLAB program (Dotfinder; [26]) manually corrected in a few cases (crosses) allowed obtaining Delaunay triangulation representations. Triangles edges shown in red possess 1 vertex on the image boundary and were ignored in the analysis. Histograms show the distributions of edge lengths for each species (y-axis: number of edges) and illustrate spacing variability, quantified as standard deviation of normalized edge lengths (see Materials and methods and Fig 2). The data underlying this figure can be found at 10.5281/zenodo.7006365. Scale bar: 500 μm.
(JPG)

**S3 Fig. Quantification of primordia size and density.** Quantifications of feather primordia size (in mm$^2$) and density (in primordia/mm$^2$) at condensation and differentiation stages are shown for each color-coded each species. Plotting spacing variability values vs. average primordia area or density at condensation stage (Cd; dots and green line; Pearson's correlation coefficient r = 0.6295 and 0.1654) and differentiation stage (Di; triangles and red line; r = 0.2865 and 0.2575) showed that spacing variability is not correlated to primordia size or density. The data underlying this figure can be found at 10.5281/zenodo.7006365. Error bars: mean with standard deviation.
(JPG)

**S4 Fig. Epidermal cell density at competence, condensation, and differentiation stages.** The 40× confocal views at epidermal levels of DAPI stains (in white) in flat embryonic skins of emu, common ostrich, domestic chicken, Japanese quail, and penguin are shown at competence stage (Co) together with corresponding schematics (black dotted squares show the position of images and see Fig 3), as well as at condensation (Cd) and differentiation (Di) stages in the inter-primordium (IP) and primordium (P) region. Quantifications of cell densities are

shown in corresponding graphs for competent stage (Co, bicolored circles primordia, in both graphs) and for condensation (Cd, dots) and differentiation (Di, circles) in primordia (left graph) and inter-primordia (right graph) regions. Cell density increased through time, inter-species variation appearing largely independent of tissue level or stage. The data underlying this figure can be found at 10.5281/zenodo.7006365. Scale bar: 20 μm. Error bars: mean with standard deviation.
(JPG)

**S5 Fig. Dermal cell density at competence, condensation, and differentiation stages.** The 40× confocal views at dermal levels of DAPI stains (in white) in flat embryonic skins of emu, common ostrich, domestic chicken, Japanese quail, and penguin are shown at competence stage (Co) together with corresponding schematics (black dotted squares show the position of images and see Fig 3), as well as at condensation (Cd) and differentiation (Di) stages in the inter-primordium (IP) and primordium (P) region. Quantifications of cell densities are shown in corresponding graphs for competent stage (Co, bicolored circles primordia, in both graphs) and for condensation (Cd, dots) and differentiation (Di, circles) in primordia (left graph) and inter-primordia (right graph) regions. Cell density increased through time, inter-species variation appearing largely independent of tissue level or stage. The data underlying this figure can be found at 10.5281/zenodo.7006365. Scale bar: 20 μm. Error bars: mean with standard deviation.
(JPG)

**S6 Fig. Epidermal cell anisotropy at competent stage.** Left panels: 40× confocal views of 100 μm$^2$ magnifications of phalloïdin stains (in white) in inter-primordia regions of the epidermis on flat embryonic skins of each species at competence stage and corresponding schematics indicating the position of images (black dotted squares) show the anisotropy of average cell shapes (as described in Fig 3; red bars). Scale bar: 20 μm. Right panel: Quantifications of anisotropy amplitude in color-coded species are represented into polar coordinates for each stage (small dots are individual values, large dots are averaged values; $n$ = 3 specimen per species). The data underlying this figure can be found at 10.5281/zenodo.7006365.
(JPG)

**S7 Fig. Epidermal anisotropy at condensation stage.** The 40× confocal views of 100 μm$^2$ magnifications of phalloïdin stains (in white) in inter-primordia (left column) and primordia (right column) regions of the epidermis on flat embryonic skins of each species at condensation stage and corresponding schematics indicating the position of images (black dotted squares) show the anisotropy of average cell shapes (as described in Fig 3; red bars). Scale bar: 20 μm. Quantifications of anisotropy amplitude in color-coded species are represented into polar coordinates for each stage (small dots are individual values, large dots are averaged values; $n$ = 3 specimen per species). The data underlying this figure can be found at 10.5281/zenodo.7006365.
(JPG)

**S8 Fig. Epidermal anisotropy at differentiation stage.** The 40× confocal views of 100 μm$^2$ magnifications of phalloïdin stains (in white) in inter-primordia (left column) and primordia (right column) regions of the epidermis on flat embryonic skins of each species at differentiation stage and corresponding schematics indicating the position of images (black dotted squares) show the anisotropy of average cell shapes (as described in Fig 3; red bars). Scale bar: 20 μm. Quantifications of anisotropy amplitude in color-coded species are represented into polar coordinates for each stage (small dots are individual values, large dots are averaged values; $n$ = 3 specimen per species). The data underlying this figure can be found at 10.5281/

zenodo.7006365.
(JPG)

**S9 Fig. Non-correlation of epidermal cell anisotropy and spacing variability.** The plot shows non-correlating averaged values of epidermal cell anisotropy at competence stage vs. spacing variability at condensation stage (black dots and green line; Pearson's correlation coefficient r = −0.4274) or at condensation stage vs. differentiation stage (circles and red line; r = 0.2543). The data underlying this figure can be found at 10.5281/zenodo.7006365. Error bars: standard deviation.
(JPG)

**S10 Fig. Dermal cell anisotropy in the primordia region in studied species.** The 40× confocal views of 100 μm$^2$ magnifications of phalloïdin stains (in white) primordia regions on flat embryonic skins of each species at condensation stage (left column) and differentiation stage (right column) and corresponding schematics indicating the position of images (black dotted squares) show the anisotropy of average cell shapes (as described in Fig 3; red bars). Scale bar: 20 μm. Quantifications of anisotropy amplitude in color-coded species are represented into polar coordinates for each stage (small dots are individual values, large dots are averaged values; $n$ = 3 specimen per species). The data underlying this figure can be found at 10.5281/zenodo.7006365.
(JPG)

**S11 Fig. Correlation of dermal anisotropy with primordia spacing variability, size, and density.** The left plot shows correlating averaged values of dermal cell anisotropy at competence stage vs. spacing variability at condensation stage (black dots and green line; Pearson's correlation coefficient r = −0.6769) or at condensation stage vs. differentiation stage (circles and red line; r = −0.6689). Middle and right plots show non-correlating values of averaged dermal cell anisotropy amplitude at competence stage vs. primordia area or density at condensation stage (black dots and green line; Pearson's correlation coefficients r = −0.84305 or −0.8039) or at condensation stage vs. differentiation stage (circles and red line; r = 0.04398 or 0.3800). The data underlying this figure can be found at 10.5281/zenodo.7006365. Error bars: standard deviation.
(JPG)

**S12 Fig. Tissue shape changes and dynamic array formation in skin explant. (A)** Dorsal skin explants of Japanese quail, emu, and African penguin embryos prepared at competence stage recapitulate timely dynamics of primordia emergence. Scale bars: 2 mm. **(B)** Quantifications of the percentage of surface area retractation at competence and differentiation stages (see Materials and methods) show that Japanese quail, emu, and penguin explants shrink by approximately 10%, 10%, and 50% of their surface, respectively. The data underlying this figure can be found at 10.5281/zenodo.7006365.
(JPG)

**S13 Fig. Primordia size and density in cultured explants. (A)** Quantifications of primordia size (in mm$^2$) at differentiation stage showed no significant difference between control flat skins and cultured explants in the Japanese quail (unpaired 2-tailed $t$ test; $p$ = 0.2776) and the emu ($p$ = 0.2180) and but a significant reduction in the penguin ($p$ < 0.0001). **(B)** Density (in primordia/mm$^2$) was conserved in the emu ($p$ = 0.0543) but significantly increased in the Japanese quail ($p$ < 0.0001) and penguin ($p$ < 0.0001). The data underlying this figure can be found at 10.5281/zenodo.7006365. Error bars: mean with standard deviation; significance of

statistical tests is shown with stars.
(JPG)

**S14 Fig. Effect of explant culture and Latrunculin A treatment in the emu. (A)** The 3,2×
(upper panel) and 40× (bottom panel) views of a phalloïdin-stained emu flat skin. **(B)** Dis-
sected portions of dorsal emu embryonic skin (i.e., explant, in blue) were cultured to differen-
tiation stage (see Fig 4). Quantifications of spacing variability show that at that stage, pattern
fidelity was maintained in explants (triangles; $n = 6$) compared to flat skis (dots; $n = 4$;
unpaired 2-tailed $t$ test; $p = 0.3963$). **(C)** Emu skin explants were treated with low-doses of
Latrunculin A (LatA, in blue) and cultured to differentiation stage (see Fig 5). Quantifications
of spacing variability showed that at this stage, low pattern fidelity was maintained (squares;
$n = 5$; $p = 0.4195$) compared to control cultured explants (triangles; $n = 5$). Confocal views and
respective quantifications (as described in Fig 3) of phalloïdin-stained inter-primordia dermis
showed that cells remain isotropic, which is typical of this species at this stage. The data under-
lying this figure can be found at 10.5281/zenodo.7006365. Error bars: mean with standard
deviation. Scale bars: 500 μm (primordia pattern), 20 μm (anisotropy).
(JPG)

**S15 Fig. Primordia array and dermal anisotropy in the African penguin.** Left panel: African
and Gentoo penguins are closely related *Neoaves* species. Middle panels: At competence stage
in both species, 10× confocal views of DAPI stains and corresponding schematics showing the
location of images in flat skins (see Fig 3; scale bars: 500 μm) and 100 μm$^2$ magnifications of
40× confocal views of phalloïdin stains in inter-primordia regions (right scale bars: 20 μm), no
pattern was visible. Right panels: At differentiation stage, African penguin skins displayed pat-
tern geometry and dermal cell anisotropy (views correspond to blue squares) identical to those
of the Gentoo penguin (and see Figs 2 and 3). Photo credits: Raphaël Sané (www.raphaelsane.
com, Gentoo penguin) and Alain Bidart (www.alainbidart.fr, African penguin).
(JPG)

**S16 Fig. Effect of Latrunculin on dermal cell density.** Quantifications of dermal cell density
normalized to 100 μm$^2$ areas at 3 different positions along the first formed row at condensation
and differentiation stages showed no significant change between control ($n = 3$) and drug-
treated ($n = 3$) Japanese quail explants in the inter-primordia region (left graph; IP; unpaired
2-tailed $t$ test, $p = 0.4076$ at condensation stage and 0.1927 at differentiation stage) and primor-
dia region (right graph; P; $p = 0.3758$ at condensation stage and 0.4895 at differentiation
stage). The data underlying this figure can be found at 10.5281/zenodo.7006365.
(JPG)

**S17 Fig. Effect of pulsed drug treatment on pattern fidelity and cell shape anisotropy.** Japa-
nese quail skin explants prepared at competence stage (in orange) were treated with low doses
of Latrunculin A (LatA, in blue) during 2 h and cultured to condensation stage (+24 h; left
panels) or differentiation stage (right panels). Quantifications of primordia spacing variability
on phalloïdin-stained control flat skins (explant; triangles) and LatA-treated cultured explants
(+LatA; squares) showed that the pulse of LatA only transiently modified pattern fidelity: it
was lower at condensation stage ($n = 8$ explants; $n = 6$ +LatA; unpaired 2-tailed $t$ test;
$p = 0.0307$) but had entirely recovered at differentiation stage ($n = 9$ explants; $n = 3$ +LatA;
unpaired 2-tailed $t$ test; $p = 0.7775$). Confocal views and respective quantifications (as
described in Fig 3) show that at condensation stage, pulsed drug treatment efficiently reduces
cell shape anisotropy compared to control explants. Small data shapes are individual values,
large data shapes (in green) are averaged values. The data underlying this figure can be found
at 10.5281/zenodo.7006365. Error bars: mean with standard deviation. Scale bars: 500 μm

(primordia pattern), 20 μm (anisotropy).
(JPG)

**S18 Fig. Effect of Latrunculin on primordia size and density.** Left graphs: Primordia size (in mm$^2$) did not change between control and drug-treated explants at differentiation stage of emus and Japanese quail (unpaired 2-tailed $t$ tests; $p = 0.2363$ and $0.1910$) but was significantly increased in drug-treated African penguin explants ($p < 0.0001$). Right graphs: Primordia density (in primordia/mm$^2$) did not change between control and drug-treated explants in the 3 species ($p = 0.9341$, $0.3369$ and $p = 0.1054$). The data underlying this figure can be found at 10.5281/zenodo.7006365. Error bars: mean with standard deviation; significance of statistical tests is shown with stars.
(JPG)

**S19 Fig. Recovery of explant retractation after stretching in penguin explants. (A)** Penguin explants were placed on collagen gels (in white) on Petri dishes (in gray) at competence stage (left panels) and cultured to reach differentiation stage (i.e., after 4 days; right panels). **(B, C)** In penguin explants prepared at competence stage (left panels) on which we applied controlled directional stretch (middle panels and see Fig 6), we observed a recovery of culture-induced retractation (right panels in B and quantifications in C; explants, white dots, $n = 4$; stretched explants, black dots, $n = 3$). The data underlying this figure can be found at 10.5281/zenodo.7006365. Scale bars: 2 mm.
(JPG)

**S20 Fig. Quantification methods for cell anisotropy, divergence, and primordia tracking. (A)** Skin explants of membrane GFP (mb-GFP) Japanese quails were cultured as described in Fig 7. Snapshots of time-lapse confocal movies on control ($n = 3$) or Latrunculin A-treated (+LatA, in blue; $n = 3$) mb-GFP skin explants were computed to produce heat maps of average cell anisotropy amplitude per interrogation box (shades-of-gray squares). Over the course of each movie, putative inter-primordia (IP; red squares) and primordia (P; orange squares) regions were defined as the 4 interrogation boxes (for evolving anisotropy amplitude; full lines) or 9 interrogation boxes (for PIV analyses; dotted lines) with respectively maximal and minimal average anisotropy (see Materials and methods). **(B)** On time-lapse movie images (1) an algorithm detecting brightest pixels, (2) then applying Gaussian smoothing (3) was used to automatically identify putative primordia (4) and track their positions along the antero-posterior axis through time (see Fig 7I–7K). In bottom panels, snapshots of a time-lapse movie show that the algorithm first automatically detected 2 primordia (black circles) at $t_{ref}$+15min, then only one at $t_{ref}$+1h15min and $t_{ref}$+4h, thereby evidencing a fusion event. **(C)** Quantifications of dermal cell anisotropy amplitude within automatically defined IP and P regions of first-formed segments showed it significantly decreased through time (in hours to $t_{ref}$, marked with a black dotted line) after drug treatment (in black) compared to control conditions (in white; unpaired 2-tailed $t$ test; $p < 0.0001$ for both IP and P). **(D)** Quantifications of the divergence of the vector field of dermal cell movement averaged within IP and P (the blue line represents value 0 at which there is no contraction or extension) showed that contraction in the primordium region, occurring 2 h after $t_{ref}$, decreases upon LatA treatment (in black) compared to control explants (in white). The data underlying this figure can be found at 10.5281/zenodo.7006365.
(JPG)

**S21 Fig. Dermal cell anisotropy in un-patterned regions. (A)** To perform time-lapse experiments in un-patterned regions (in yellow), dorsal skin explants of membrane-GFP (mb-GFP) Japanese quails were placed dermal side down on nitrocellulose filters for confocal imaging

(see Materials and methods). **(B)** Snapshots at t0 (and corresponding schematic), t+1 and t+2h of a time-lapse confocal movie in the un-patterned region. **(C)** Quantifications of cell shape anisotropy as described in Fig 3 show that dermal cells in the un-patterned region are isotropic. The data underlying this figure can be found at 10.5281/zenodo.7006365. Scale bar: 50 μm. Small dots are individual values; large dots (in green) are averaged values.
(JPG)

**S1 Movie. A 2 h time-lapse confocal movie in the un-patterned region of a membrane-GFP Japanese quail showed random and slow dermal cell movement.**
(AVI)

**S2 Movie. An 8 h time-lapse confocal movie (movie #2 in Fig 7J) of a membrane-GFP Japanese quail dorsal skin showed the dynamic emergence of 2 primordia at $t_{ref}$.**
(AVI)

**S3 Movie. An 8 h time-lapse confocal movie (movie #2 in Fig 7K) of a Latrunculin A-treated membrane-GFP Japanese quail dorsal skin showed the dynamic emergence of 2 primordia at $t_{ref}$.**
(AVI)

## Acknowledgments

We thank J. Gros and members of his laboratory for providing membrane-GFP Japanese quail fertilized eggs, as well as E. Bouvet and M. Ladner for their care of the zebra finch breeding colony. We thank C. Longtine for help with Gentoo penguin harvesting in the field, H. and M.P Delignières at Dunbar Farm and L. Clifton for providing landowner permission and accommodation for fieldwork at Grave Cove, Stevely Bay, and Weddell Island, and Denise Blake, Nick Rendell, and the Falkland Islands Government for help with collection permits. We thank Ponant, Oceanwide Expeditions, all expedition staff and crew on-board, and A. Liddle for logistical help and support in assessing field sites. We thank T. Petit and the staff of the La Palmyre Zoo for their continuous help with supplying African penguin eggs. We thank the Collège de France imaging facility staff for help with time-lapse experiments and S. Chanet for helpful comments on the manuscript.

## Author Contributions

**Conceptualization:** Marie Manceau.

**Data curation:** Camille Curantz, Richard Bailleul, Magdalena Hidalgo, Melina Durande, François Graner, Marie Manceau.

**Formal analysis:** Camille Curantz, Richard Bailleul, María Castro-Scherianz, Magdalena Hidalgo, François Graner, Marie Manceau.

**Funding acquisition:** Marie Manceau.

**Investigation:** Camille Curantz, Richard Bailleul, María Castro-Scherianz, Magdalena Hidalgo, Marie Manceau.

**Methodology:** Camille Curantz, Richard Bailleul, María Castro-Scherianz, Magdalena Hidalgo, Melina Durande, François Graner.

**Project administration:** Marie Manceau.

**Resources:** Marie Manceau.

**Software:** Richard Bailleul, Melina Durande, François Graner.

**Supervision:** Marie Manceau.

**Validation:** Camille Curantz, Magdalena Hidalgo, Marie Manceau.

**Visualization:** Magdalena Hidalgo, Marie Manceau.

**Writing – original draft:** Camille Curantz, Magdalena Hidalgo, Marie Manceau.

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
