## [Editor Report · Decision Letter 0]

6 Feb 2022

Dear Dr Manceau, 

Thank you for submitting your manuscript entitled "Cell shape anisotropy constrains self-organized pattern fidelity in birds" for consideration as a Research Article by PLOS Biology.

Your manuscript has now been evaluated by the PLOS Biology editorial staff as well as by an academic editor with relevant expertise and I am writing to let you know that we would like to send your submission out for external peer review.

Once your full submission is complete, your paper will undergo a series of checks in preparation for peer review. Once your manuscript has passed the checks it will be sent out for review. To provide the metadata for your submission, please Login to Editorial Manager (https://www.editorialmanager.com/pbiology) within two working days, i.e. by Feb 08 2022 11:59PM.

If your manuscript has been previously reviewed at another journal, PLOS Biology is willing to work with those reviews in order to avoid re-starting the process. Submission of the previous reviews is entirely optional and our ability to use them effectively will depend on the willingness of the previous journal to confirm the content of the reports and share the reviewer identities. Please note that we reserve the right to invite additional reviewers if we consider that additional/independent reviewers are needed, although we aim to avoid this as far as possible. In our experience, working with previous reviews does save time. 

If you would like to send previous reviewer reports to us, please email me at ialvarez-garcia@plos.org to let me know, including the name of the previous journal and the manuscript ID the study was given, as well as attaching a point-by-point response to reviewers that details how you have or plan to address the reviewers' concerns. 

Given the disruptions resulting from the ongoing COVID-19 pandemic, please expect some delays in the editorial process. We apologise in advance for any inconvenience caused and will do our best to minimize impact as far as possible.

Kind regards,

Ines

--

Ines Alvarez-Garcia, PhD

Senior Editor

PLOS Biology

---

## [Decision Letter · Decision Letter 1]

1 Apr 2022

Dear Dr Manceau,

Thank you for submitting your manuscript entitled "Cell shape anisotropy constrains self-organized pattern fidelity in birds" for consideration as a Research Article at PLOS Biology. Thank you also for your patience as we completed our editorial process, and please accept my apologies for the delay in providing you with our decision. Your manuscript has been evaluated by the PLOS Biology editors, an Academic Editor with relevant expertise, and by two independent reviewers.

As you will see, the reviewers find your conclusions interesting and worth pursuing, but they also raise several points that they will need to be addressed to strengthen the conclusions. Reviewer 1 proposes some experiments to show that stretch-induced anisotropy reduces motility in order to confirm the model proposed and to decouple anisotropy from motility to demonstrate causal relationship between constrained vs unconstrained movement and ordered vs disordered primordial array. This reviewer also thinks you should clarify some apparent contradictions in the model and improve the figures, whereas Reviewer 2 feels that the background context of the work could be improved and that some of the experiments need to be quantified.

In light of the reviews (attached below), we will not be able to accept the current version of the manuscript, but we would welcome re-submission of a revised version that takes into account the reviewers' comments. We cannot make any decision about publication until we have seen the revised manuscript and your response to the reviewers' comments. Your revised manuscript is also likely to be sent for further evaluation by the reviewers.

We expect to receive your revised manuscript within 3 months. 

**IMPORTANT - SUBMITTING YOUR REVISION**

3. Resubmission Checklist

a) *PLOS Data Policy*

b) *Published Peer Review*

c) *Blurb*

Please also provide a blurb which (if accepted) will be included in our weekly and monthly Electronic Table of Contents, sent out to readers of PLOS Biology, and may be used to promote your article in social media. The blurb should be about 30-40 words long and is subject to editorial changes. It should, without exaggeration, entice people to read your manuscript. It should not be redundant with the title and should not contain acronyms or abbreviations. For examples, view our author guidelines: https://journals.plos.org/plosbiology/s/revising-your-manuscript#loc-blurb

Sincerely,

Ines

--

Ines Alvarez-Garcia, PhD

Senior Editor

PLOS Biology

Reviewers' comments

Rev. 1:

Spatial arrangements of cells into patterns often arise when stochastic changes in cell properties are stabilized and amplified. It is therefore unclear how reproducible self-organization, which is crucial to organogenesis and organism survival, is achieved across individuals and species. A key property in tissues undergoing patterning is tissue anisotropy. This study uses a cross-species comparative approach to address the role of dermal anisotropy during avian feather follicle patterning. The authors first show that birds of distinct clades that display variable regularity of their feather primordia arrays also have corresponding differences in anisotropy--higher anisotropy generally correlates, in time and between species, with the appearance of more regular patterns. The authors perturbed anisotropy pharmacologically and mechanically, and saw that a shift toward isotropic organization using actin polymerization inhibitor LatA and uniaxial stretch orthogonal to existing tissue orientation caused more disordered follicle patterns. Conversely, forced anisotropic organization using stretch restored pattern order in a disordered follicle pattern/isotropic dermis condition. Finally, live imaging showed that LatA caused higher cell motility and destabilized the shapes and dynamics of primordia condensates. Altogether the authors propose a model where tissue anisotropy constrains cell motility and limits the positioning of primordia into ordered patterns, while isotropic arrangement allows for free movement into randomized arrays.

This study adds to a growing body of knowledge concerning physical mechanisms of spatial pattern morphogenesis. The authors define an intriguing and somewhat unexpected relationship between anisotropy and motility on the scale of individual cells to explain a tissue-level phenomenon of pattern fidelity. This is an important contribution, as links between tissue and cell-scale physical processes often remain poorly defined as we learn more about the role of mechanics in development. Furthermore, because the authors take advantage of species-specific differences, findings may be considered in an evolutionary context, and could inform potential functional consequences of feather patterning for avian survival and adaptation. On a technical level, the authors use a variety of cross-disciplinary techniques and robust quantifications with many replicates to support their findings. While this submission is certainly appropriate for PLoS Biology, being an interesting study, and potentially making a significant contribution to the field, several of the conclusions need to be better substantiated prior to acceptance:

Major Critiques:

1. The part of the model concerning cell motility comes only from an experiment using Latruncilin A, which inhibits actin polymerization and presumably directly affects migratory properties. Even though the LatA effect on cells is the opposite of what one would expect (lower motility), it is unclear whether isotropic organization and high motility are independent downstream effects of perturbing actin, rather than motility depending on cell shape. Experiments demonstrating that stretch-induced anisotropy reduces motility, and/or that cells in isotropic dermal tissues from paleognaths are more motile than in quails or chickens, are needed to lend credibility to this part of the model.

2. The proposed causal relationship between constrained vs unconstrained movement and an ordered vs. disordered primordial array (respectively) is not well demonstrated in this study alone. The addition of simple 2D mathematical model or, at the very least, references to established works that support this effect of motility on pattern order is needed. Additional experiments decoupling anisotropy from motility in the system would also strengthen the argument, though these would be understandably difficult to do.

3. There are a few results that seemingly contradict the notion that anisotropy leads to less spacing variability in the primordia array (and vice versa), which is central to the paper's findings. 1) Comparing Fig. 3B and C (emu vs quail), the distributions and means of amplitudes are very similar, but the spacing variabilities appear different between species—as different as penguin flat vs explant in Fig. 3D. 2) In Fig. 3D, the penguin explant primordia pattern shows higher spacing variability, but the cell arrangement still appears anisotropic, just oriented orthogonally to the penguin flat skin in Fig. 2 (penguin explant appears nearly as AP oriented as the Fig. 2 Cd stage Japanese quail). 2) In Fig. 5C, though apparently not significant, the DV+48hr amplitudes appear more DV oriented than the control, but have significantly more spacing variability. Though the authors somewhat acknowledge these findings, they should include a more complete discussion/justification in the text of how these apparent contradictions affect the final model.

Minor Critiques:

1. Some stained fluorescent images are either too small or too dim, eg. Fig. 1B, Fig. 2A, Fig. S5

2. Fig. 3-5, S12 individual data points in anisotropy amplitude plots are way too small (can't distinguish shapes). This may be a personal preference but the shapes are inconsistent, eg. explant is triangle in Fig. 3 but control is triangle in Fig. 4.

3. Fig. 6, S14 black and white curves and data points need a legend somewhere specifying explant/explant + LatA. Fig 5D box plots need x-axis labels.

4. Fig. 6B,C plots of Primordia position vs. Time are somewhat hard to interpret. Perhaps something like distance between primordia or primordia aspect ratio over time would be more clear?

Rev. 2:

Feather primordia form exquisite patterns. In emu, chicken, finch, penguin, etc. there are differences in the bud size, spacing, the way they form (propagation versus simultaneously), yet in each case the basic periodic feather patterns form. Therefore, the authors ask an interesting question on how the pattern fidelity in the dorsal skin of birds are maintained and hoping to use this approach to decipher the fundamental principles of feather pattern formation. In this manuscript, then then focus on the roles of cell shape anisotropy and hypothesize that it is important for fidelity of feather pattern formation. They did some substantial analyses based on this hypothesis. This comparative pattern formation study is interesting: they compare cell shapes and the emergence of feather patterns. Some whole mount molecular expression figures are beautiful, but some are unclear. The writing is dense and should be improved to make it clearer for general readers to follow. Overall, the conclusion is cell shape anisotropy can constrain avian dorsal feather pattern fidelity. Yet, many parameters can affect cell shape anisotropy, and these are not explored. The citation and discussion of the literature need improvement. Many papers directly relevant to this manuscript are not cited properly and discussed - in fact some findings in these literatures are in line with the concept here. So, it is good for the whole field if authors can integrate them and make a well-synthesized discussion. There are also gaps which should be addressed.

Major comments:

1. About periodic formation of feather germs, pattern fidelity:

In an earlier paper, it is shown that when dissociated chicken dermal cells are recombined with epithelium, the dissociate dermal cells mediate simultaneous formation of feather primordia. These newly formed buds are always of the same size. But the bud number varies, and is the function of the number of dermal cells added. With the increase bud density, the inter-bud spacing decreases. Although "pattern fidelity", "cell shape anisotropy" is novel in this manuscript, this classical paper is a beautiful example of converting a pattern with fidelity in the chicken in vivo to a non-fidelity pattern in vitro due to the disruption of dermal cell anisotropy (and something else). The work is in line with the authors' model and should be cited and discussed together.

Jiang, T. X., Jung, H. S., Widelitz, R. B., Chuong, C. M. (1999). Self-organization of periodic patterns by dissociated feather mesenchymal cells and the regulation of size, number and spacing of primordia. Development, 126(22)4997-5009.

2. Cell shape anisotropy. This term refers to cell properties at the morphological level. Proliferation control, cell migration, etc. It is ok to focus the question on this level, where one might expect a global level cell shape anisotropy and local level cell shape anisotropy. Many biophysical and biochemical factors can affect cell shape anisotropy. Please discuss the inputs of these underlying factors regulating cell shape more clearly. In addition to Eda, FGF 20 (Headon, PLOS bio) which was cited, Lin et al., 2009 also discussed this issue and should be cited.

Lin, C. M., Jiang, T. X., Baker, R. E., Maini, P. K., Widelitz, R. B., Chuong, C. M. (2009). Spots and stripes: Pleomorphic patterning of stem cells via p-erk-dependent cell chemotaxis shown by feather morphogenesis and mathematical simulation. Dev Biol, 334(2)369-382.

3. About bud forming simultaneously or progressively:

Chuong group has explained their view in Inaba et al., 2019. It stated Turing patterning can occur locally, if threshold is achieved (as seen in Emu). Or, global events can add to lower the threshold that triggers Turing patterning. Cell shape anisotropy could be part of the unknown global events mentioned in the Inaba et al paper. While these authors may have other views. Whatever their views are, the concept of this relevant paper should be discussed and reconciled in the discussion.

Inaba M, Harn HI, Chuong CM. Turing patterning with and without a global wave. PLoS Biol. 2019 17:e3000195.

4. In Figure 3, in ex vivo cultures, primordia density increased and primordia size decreased, consistent with the marked reduction in explant size in this species. The data for spacing variability was shown. Please provide further quantification of the decreased explant size. Also, please explain more clearly why this led to increased feather numbers with smaller size.

Minor comments

1. Title: probably better to specify feather to be clearer what this paper is about. "Feather pattern fidelity" instead of "pattern fidelity".

2. Fig. 1B and Fig. S1

- The orientation of Fig. 1B and Fig. S1 should be clearly marked.

- How the placement of the broken white line drawn was decided is not clear. There does not appear to be a clear border and the line can be arbitrary. Perhaps they can draw a half a circle on one figure and leave the other to show there is a boundary.

- White arrowheads point to beta-catenin. It is asymmetrical. Is this asymmetry position consistent in all buds?

3. Fig.1D, the two right columns, the position of the midline should be indicated and the number of rows away from the midline should be stated. Beta-catenin and bud sizes are the same in many birds, but different in emu. Why?

In penguin, in Co stage, beta-catenin appears to be a circle, and in Cd stage, beta-catenin appears to be on one side only?

4. Fig. 2. and Fig. S5, since the study is about the shape, the position of the midline, and the position of the panel within the body (how many rows lateral to the midline) should be marked clearly in a simple schematic drawing. Within the panel, the location of the bud should also be pointed out. Or do the red bars mark the boundary of the primordia?

If the midline is the same for all, the dermal cells appear to be anterior-posterior oriented in the chicken, but medial-lateral oriented toward the midline in the penguin. But the skin from both birds form buds. Is this true?

5. Fig. S1. Are all buds from the same position of the embryos? Why Zebra Finch shows a different configuration in Cd specimens? I understand it is not easy to get all specimens to be in the same positions. But this has to be made clear so the findings can be useful to others.

6. Fig. S4, It is valuable to have the cell density data. Again, the orientation of the panel in the context of the whole embryo has to be accompanied by a schematic drawing.

7. Fig. S8. The two columns under Co. Are they of the same orientation? Is this what you are trying to show with the red bars?

8. In introduction and discussion, the following papers should be properly cited.

About the role of BMP, FGF in Turing periodic patterning of feathers, the first paper on this topic should be cited.

Jung, H. S., Francis-West, P. H., Widelitz, R. B., Jiang, T. X., Ting-Berreth, S., Tickle, C., Wolpert, L., Chuong, C. M. (1998). Local inhibitory action of Bmps and their relationships with activators in feather formation: Implications for periodic patterning. Dev Biol, 196(1)11-23.

About whole mount beta-catenin in situ pattern and the shift of bud position during patterning:

Chuong, C. M., Yeh, C. Y., Jiang, T. X., Widelitz, R. B. (2013). Module based complexity formation: Periodic patterning in feathers and hairs. Wiley interdisciplinary reviews Developmental biology, 2(1)97-112.

---

## [Decision Letter · Decision Letter 2]

5 Aug 2022

Dear Dr Manceau,

Thank you for your patience while we considered your revised manuscript "Cell shape anisotropy constrains self-organized feather pattern fidelity in birds" for publication as a Research Article at PLOS Biology. This revised version of your manuscript has been evaluated by the PLOS Biology editors, the Academic Editor and the original reviewers.

Based on the reviews, we are likely to accept this manuscript for publication, provided you satisfactorily address the remaining points raised by the reviewers. Please also make sure to address the following data and other policy-related requests.

IMPORTANT:

a) Please address the remaining requests from reviewer #2.

b) We agree with this reviewer's suggestion that it would be appropriate to replace "constrains" with "contributes to" in the Title.

c) Please address my Data Policy requests below; specifically, we need you to supply the numerical values underlying Figs 2C, 3 (bottom panels), 4CEIN, 5BCEF, 6CEFHI, 7FGHJK, S2 (right panels), S3, S4 (bottom panels), S5 (bottom panels), S6 (right panel), S7 (bottom panels), S8 (bottom panels), S9, S10 (bottom panels), S11, S12B, S13AB, S14BC, S16, S17, S18, S19C, S20CD, S21C, either as a supplementary data file or as a permanent DOI’d deposition like Zenodo, Dryad, Figshare, etc..

d) Please cite the location of the data clearly in all relevant main and supplementary Figure legends, e.g. “The data underlying this Figure can be found in S1 Data.”

We expect to receive your revised manuscript within two weeks. 

*Published Peer Review History*

*Press*

Sincerely,

Roli Roberts

Roland G Roberts PhD

Senior Editor

PLOS Biology

rroberts@plos.org

on behalf of

Ines Alvarez-Garcia, PhD

Senior Editor,

ialvarez-garcia@plos.org,

PLOS Biology

DATA POLICY:

Regardless of the method selected, please ensure that you provide the individual numerical values that underlie the summary data displayed in the following figure panels as they are essential for readers to assess your analysis and to reproduce it: Figs 2C, 3 (bottom panels), 4CEIN, 5BCEF, 6CEFHI, 7FGHJK, S2 (right panels), S3, S4 (bottom panels), S5 (bottom panels), S6 (right panel), S7 (bottom panels), S8 (bottom panels), S9, S10 (bottom panels), S11, S12B, S13AB, S14BC, S16, S17, S18, S19C, S20CD, S21C. NOTE: the numerical data provided should include all replicates AND the way in which the plotted mean and errors were derived (it should not present only the mean/average values).

SPECIES INDICATED IN THE ABSTRACT? 

- Please note that per journal policy, the model system/species studied should be clearly stated in the abstract of your manuscript. 

We require the original, uncropped and minimally adjusted images supporting all blot and gel results reported in an article's figures or Supporting Information files. We will require these files before a manuscript can be accepted so please prepare and upload them now. Please carefully read our guidelines for how to prepare and upload this data: https://journals.plos.org/plosbiology/s/figures#loc-blot-and-gel-reporting-requirements

DATA NOT SHOWN?

REVIEWERS' COMMENTS:

Reviewer #1:

The authors have done a very nice job addressing my concerns, and I am happy to recommend publication of this revised submission in its current form.

Reviewer #2:

The revision has improved and many previous concerns are addressed properly. Authors ask an interesting question in an unexplored topic and has accumulated interesting observation to demonstrate a correlation between cell shape anisotropy and feather pattern fidelity. 

Major: 

Yet, a major concern I share with R1 is that the causality of cell shape anisotropy and pattern fidelity is not strongly established. Authors did try to test the relationship. But the use of latrunculin and biostretch are not super specific and they state they are not able to evaluate the mechanism at cellular (e.g., cell motility as asked by R1) or molecular level in the allowed time frame. Both latrunculin and biostretch may cause changes other than cell shape anisotropy. Thus, while the work is still beautiful and valuable, I suggest authors tune down the claim and acknowledge the possibility of other mechanisms. For example, in the title, authors may consider replace "constrains" with "contributes to" . It is good for the community and authors to leave some spaces for other mechanisms. Also, in the text, please adjust some wording. For example, line 168, "influence" is too strong. Also, please similar type of wording in other places. 

167 These observations suggested that

168 emerging properties of cell anisotropy influence the self-organization of primordia such that

169 they result in arrays with species-specific fidelity.

Minor:

Fig. 1. Panel C and D, the positions of the primordia. I suppose they are lined up anterior-posteriorly. Where is the midline? Please mark them. Also, in epidermis, in condensation and differentiation stage, why strong beta catenin staining is in the posterior and anterior primordia respectively?

Fig. 2C, Cd, Di should be defined in figure legends here. On the differentiation panel of emu and ostrich, why the primordia are different in size between beta catenin and DAPI staining?

---

## [Editor Report · Decision Letter 3]

26 Aug 2022

Dear Dr Manceau,

Thank you for the submission of your revised Research Article entitled "Cell shape anisotropy contributes to self-organized feather pattern fidelity in birds" for publication in PLOS Biology. On behalf of my colleagues and the Academic Editor, Marianne Bronner, I am happy to say that we can in principle accept your manuscript for publication, provided you address any remaining formatting and reporting issues. These will be detailed in an email you should receive within 2-3 business days from our colleagues in the journal operations team; no action is required from you until then. Please note that we will not be able to formally accept your manuscript and schedule it for publication until you have completed any requested changes.

PRESS

Sincerely, 

Ines

--

Ines Alvarez-Garcia, PhD

Senior Editor

PLOS Biology
